# Attribution of ground-level ozone to anthropogenic and natural sources of $NO_x$ and reactive carbon in a global chemical transport model

Tim Butler[1,2], Aurelia Lupascu[1], and Aditya Nalam[1,2]

[1]Institute for Advanced Sustainability Studies, Potsdam, Germany
[2]Institut für Meteorologie, Freie Universität Berlin, Germany

**Correspondence:** Tim Butler (tim.butler@iass-potsdam.de)

**Abstract.** We perform a source attribution for tropospheric and ground-level ozone using a novel technique which accounts separately for the contributions of the two chemically distinct emitted precursors (reactive carbon and oxides of nitrogen) to the chemical production of ozone in the troposphere. By tagging anthropogenic emissions of these precursors according to the geographical region from which they are emitted, we determine source/receptor relationships for ground-level ozone.

Our methodology reproduces earlier results obtained through other techniques for ozone source attribution, and also delivers additional information about the modelled processes responsible for intercontinental transport of ozone, which is especially strong during the spring months. The current generation of chemical transport models used to support international negotiations aimed at reducing the intercontinental transport of ozone show especially strong inter-model differences in simulated springtime ozone. Current models also simulate a large range of different responses of surface ozone to methane, one of the

major precursors of ground-level ozone. Using our novel source attribution technique, we show that emissions of $NO_x$ from international shipping over the high seas play a disproportionately strong role in our model system to the hemispheric-scale response of surface ozone to changes in methane, as well as to the springtime maximum in intercontinental transport of ozone and its precursors. We recommend a renewed focus on improvement of the representation of the chemistry of ship $NO_x$ emissions in current-generation models. We demonstrate the utility of ozone source attribution as a powerful model diagnostic tool,

and recommend that similar source attribution techniques become a standard part of future model inter-comparison studies.

## 1 Introduction

Tropospheric ozone plays a central role in the chemistry and self-cleansing capacity of the troposphere (Crutzen, 1973; Monks et al., 2015), but at high concentrations close to the ground, it is harmful to human health (Haagen-Smit, 1952; Fleming et al., 2018) and vegetation (Reich and Amundson, 1985; Mills et al., 2018). As well as being transported into the troposphere

through exchange with the stratosphere, ozone can be formed through chemical reactions in the troposphere involving two chemically distinct precursors: oxides of nitrogen (collectively $NO_x$); and reactive carbon species, including carbon monixide, methane, and volatile organic compounds (Crutzen, 1973; Atkinson, 2000). Increases in tropospheric ozone since preindustrial

times have been attributed primarily to increases in anthropogenic emissions of $NO_x$ and methane, the most abundant reactive carbon species in the atmosphere (Wang and Jacob, 1998; Stevenson et al., 2013).

Ozone is long-lived enough in the troposphere to circumnavigate the Northern Hemisphere along the prevailing westerly winds (Jacob et al., 1999). Emissions of $NO_x$ or reactive carbon in any Northern Hemisphere source region can thus contribute to the ozone mixing ratio in any other region of the Northern Hemisphere. This long-range contribution to the ozone mixing ratio is often referred to as "baseline" ozone (Parrish et al., 2017; Derwent et al., 2018). Due to seasonal variation in the lifetime of ozone, this effect is strongest in spring and weakest in summer (Fiore et al., 2009). The ambient ozone mixing ratio at any

location is a combination of ozone transported from the hemispheric background, and in-situ photochemical production. Recent analyses of long-term trends in baseline ozone in western regions of North America (Parrish et al., 2017) and Europe (Derwent et al., 2018) have shown increasing trends since reliable measurements began in the 1980s until approximately 2000-2010, and indicate that these trends may be beginning to reverse.

    Chemical Transport Models (CTMs) are commonly used to interpret observations of ozone, and synthesise understanding

of the fundamental processes controlling its origin and fate in the atmosphere in order to project future trends (eg. Young et al., 2018). The range in values of surface ozone mixing ratio over the Northern Hemisphere simulated by contemporary CTMs is extremely high (see for example our Figure 1 in Section 3), requiring the use of a large ensemble of models (eg. HTAP, 2010; Young et al., 2018). When compared with available measurements of ozone for the Northern Hemisphere (eg. Schultz et al., 2017), ensembles of global CTMs are generally able to simulate the spatial distribution and seasonal cycles of surface ozone,

but are consistently biased high in the Northern Hemisphere, and have difficulty in simulating long-term trends (Young et al., 2018). Potential sources of uncertainty in CTMs include uncertainties in their chemical mechanisms (the representations of the relevant chemical reactions and their rates), their representation of atmospheric transport processes, as well as exchange processes between the atmosphere and the surface of the Earth, including emissions of the ozone precursors $NO_x$ and reactive carbon.

The most important class of reactions for the formation of ozone in the troposphere is the reaction of NO (nitric oxide) with a peroxy radical, which is itself formed during the oxidation of reactive carbon (Atkinson, 2000). During this process, the NO is converted to $NO_2$ (nitrogen dioxide), which can be rapidly photolysed, ultimately forming ozone and recycling NO. The ozone production efficiency of $NO_x$ (the combined concentration of NO and $NO_2$) can vary significantly depending on the location and timing of the $NO_x$ emissions. In the polluted boundary layer, $NO_x$ is rapidly removed from the atmosphere through the

reaction of $NO_2$ with OH, forming $HNO_3$, which is subsequently lost via dry or wet deposition. Under less polluted conditions, $NO_2$ photolysis competes more effectively with $HNO_3$ production, allowing each unit of $NO_x$ to react with a higher number of peroxy radicals before eventually being scavenged by OH, thus leading to higher ozone production efficiency per unit of $NO_x$. When $NO_x$ is lofted into the free troposphere, its ozone productivity increases substantially (Jacob et al., 1996). Emissions of $NO_x$ in the tropics are thus especially effective at producing tropospheric ozone due to being transported aloft due to

deep convection (Zhang et al., 2016). $NO_x$ emissions from both aircraft and lightning are also highly efficient at producing tropospheric ozone (Beck et al., 1992; Dahlmann et al., 2011). Combustion of fossil fuels is the largest source of $NO_x$ in the atmosphere (Galloway et al., 2008).

Lawrence and Crutzen (1999) first pointed out that international shipping can have a disproportionately high influence on tropospheric ozone due to the disperse nature of $NO_x$ emissions from this source. Hoor et al. (2009) quantified the sensitivity of tropospheric ozone to $NO_x$ emissions from different modes of transport (land, sea, and air), finding that aircraft emissions were most efficient at producing ozone (per molecule of $NO_x$ emitted), followed by ships and then land transport. For near-surface ozone however, $NO_x$ emissions from ships were shown to have a higher influence than $NO_x$ emissions from aircraft. Ozone production from ship $NO_x$ is however highly uncertain in current CTMs. Kasibhatla et al. (2000) and von Glasow et al. (2003) showed that global CTMs, due to their coarse resolution (usually in the hundreds of km) do not resolve the chemistry of ship exhaust plumes, which tends to remove $NO_x$ from the atmosphere more quickly than simulated by the global CTMs, which effectively instantly dilute these emissions into very large volumes. Wild and Prather (2006) also showed that this effect applies more generally to other concentrated emission sources such as urban areas. Vinken et al. (2011) introduced a method for parameterising ship exhaust plume chemistry using lookup tables in their global CTM, but this method has not been widely adopted by the modelling community. Modelling of ship $NO_x$ and its effects on the atmosphere remains a challenge for global CTMs.

The term "reactive carbon" encompasses a very wide range of atmospheric constituents (eg. Chameides et al., 1992; Goldstein and Galbally, 2007; Heald and Kroll, 2020). In contrast to $NO_x$, most of the reactive carbon emitted to the atmosphere is not of anthropogenic origin, but rather emitted from the biosphere. In this study we restrict our definition to molecules which yield peroxy radicals (either hydroperoxy radicals $HO_2$ or organic peroxy radicals $RO_2$) during their gas phase oxidation, and thus contribute to ozone formation by potentially converting NO to $NO_2$. This definition thus includes carbon monoxide (CO) and the large family of molecules known as Volatile Organic Compounds (VOC). The simplest VOC is methane, which is often considered separately from Non-Methane Volatile Organic Compounds (NMVOC) due to its very long lifetime in the troposphere. The ozone production potential of reactive carbon depends on the rate at which it is oxidised in the atmosphere, usually through reaction with the OH radical (Carter, 1994), as well as the subsequent chemistry of its oxidation products (Butler et al., 2011; Derwent, 2020). Most reactive carbon species have relatively short lifetimes in the troposphere due to their reaction with OH radicals. Methane, due to its exceptionally low reactivity is well-mixed in the troposphere. In contrast to other forms of reactive carbon, emissions of methane can contribute to ozone formation at any location in the troposphere where photochemical conditions are favourable (Fiore et al., 2008). Despite its low reactivity in comparison to other types of reactive carbon, methane is highly abundant, and has been shown to make a large contribution to tropospheric ozone (Wang and Jacob, 1998; Fiore et al., 2008; Stevenson et al., 2013; Butler et al., 2018).

PAN (Peroxyacetyl Nitrate) is an important reservoir species for both peroxy radicals and for $NO_x$ (Fischer et al., 2014). Peroxyacetyl radicals are formed during the oxidation of a wide range of different types of NMVOC from a wide range of different sources. PAN is formed through reaction of peroxyacetyl radicals with $NO_2$, primarily in the polluted boundary layer where both are abundant (Atkinson, 2000). The lifetime of PAN is strongly temperature dependent. At colder temperatures higher in the troposphere, PAN can be transported over long distances, and act as a source of $NO_2$ and peroxyacetyl radicals in remote regions upon subsidence and thermal decomposition (Fischer et al., 2014). The chemical mechanisms and reaction rate constants involved in the formation and decomposition of PAN vary widely between CTMs (Emmerson and Evans, 2009;

Knote et al., 2015), leading to large inter-model differences in simulated PAN (Emmons et al., 2015). Fiore et al. (2018) has suggested that measurements of PAN at northern midlatitude mountaintop sites in spring could provide a useful constraint on CTMs, although the number of observations available is limited.

With careful interpretation, the results of ensembles of CTMs can be used to diagnose long-range transboundary transport of ozone, and to develop intercontinental source-receptor relationships, relating the effects of precursor emissions from different regions of the Northern Hemisphere to mixing ratios of ground-level ozone in other regions of the Northern Hemisphere. An example is the activity of the Task Force on Hemispheric Transport of Air Pollution (HTAP, 2010), which reports to the Convention on Long-Range Transboundary Air Pollution (CLRTAP) and thus informs international policymaking for the mitigation of air pollution. The TF-HTAP studies used a "perturbation" approach, in which a control simulation was compared with sensitivity simulations in which emissions of particular ozone precursors were reduced by 20%. Combined 20% reductions of global average methane and remote anthropogenic emissions of $NO_x$, CO, and NMVOC were shown to have an approximately equal effect on annual average ozone as 20% reduction of local precursor emissions, indicating the strong role of long-range transport in influencing surface ozone in the Northern Hemisphere. Results derived from the phase one of the TF-HTAP exercise (and the phase two exercise described by Galmarini et al., 2017) are discussed in more detail by Fiore et al. (2009); Reidmiller et al. (2009); Huang et al. (2017); Jonson et al. (2018).

An alternative approach to the perturbation technique for source attribution is "tagging" (eg. Wang et al., 1998; Dunker et al., 2002; Grewe et al., 2010; Emmons et al., 2012; Derwent et al., 2015; Grewe et al., 2017; Butler et al., 2018; Bates and Jacob, 2020). When applied to ozone source attribution in a CTM, this technique involves labelling (or "tagging") modelled ozone with the identity of either the geographical region in which it is chemically produced, or with the identity of the emitted precursor(s) which ultimately led to its production. A common challenge faced by all tagging approaches is that the production of one molecule of ozone in the troposphere requires two precursors: one molecule of NO; and one peroxy radical (produced during reactive carbon oxidation). Should the ozone molecule inherit its tag from the emitted $NO_x$, the emitted reactive carbon, or in some other way? Butler et al. (2018) provides a detailed review of the different approaches to answering this question, including the trade-offs made in each case. Butler et al. (2018) also describe a novel and unique tagging methodology which allows separate attribution of tropospheric ozone to both its $NO_x$ and its reactive carbon precursor, at the cost of extra computational expense compared with other tagging methodologies. Recent work from Bates and Jacob (2020) takes the approach of defining an extended odd oxygen family including peroxy radicals, which effectively shifts the production of odd oxygen purely to photolysis reactions. Further comparison of different tagging approaches is beyond the scope of this manuscript, but remains an interesting topic for future work.

Tagging and perturbation approaches are complementary to each other (Clappier et al., 2017; Thunis et al., 2019; Mertens et al., 2020). While tagging delivers information about the contribution of different emission sources to a pollutant of interest, perturbation studies deliver information about the sensitivity of pollutants to changes in emissions, including changes in the chemical lifetimes of pollutants in response to changes in emissions. In the absence of nonlinear chemical interactions, these two different approaches ultimately yield the same results, but for tropospheric ozone, which can under some circumstances show highly nonlinear interactions between its $NO_x$ and reactive carbon precursors, these approaches can sometimes yield

very different results (Grewe et al., 2010; Mertens et al., 2018). Since air pollution mitigation strategies must involve some change in emissions, perturbation studies will always be necessary for policy-relevant modelling of atmospheric chemistry.

However, tagging studies on their own can play a role in helping to identify which emissions to mitigate (Grewe et al., 2010). When combined with perturbation studies, tagging can reveal how the contribution of unmitigated sources to ozone changes in response to mitigation measures (Mertens et al., 2018). Butler et al. (2018) have also noted that tagging studies provide useful diagnostic information about model processes, and argued for their inclusion in model inter-comparison exercises. The method described by Butler et al. (2018) is currently the only available approach which provides separate attribution of tropospheric

ozone to its $NO_x$ and reactive carbon precursors. Other schemes take different approaches to the attribution of ozone to these two chemically distinct precursors. A thorough review of several different approaches is presented in Butler et al. (2018).

In this study we use the ozone tagging methodology previously described by Butler et al. (2018) to perform a source attribution for ground-level ozone to both $NO_x$ and reactive carbon. This work builds on the work of Butler et al. (2018) by tagging anthropogenic emissions of $NO_x$ and reactive carbon by their geographical source region, and examining the seasonal cycle of

140 the surface ozone attribution in receptor regions as defined in the HTAP Phase 2 exercise. By performing separate attribution of ground-level ozone to both $NO_x$ and reactive carbon (including methane), we hope to provide more useful information to inform emission mitigation scenarios. We also show how our tagging methodology can be used as a model diagnostic tool to understand the atmospheric budgets of ozone and PAN in more detail than previously possible, potentially informing efforts to reduce the currently high level of inter-model uncertainty. Furthermore, we examine the changing contributions of the different

sources of $NO_x$ and reactive carbon to a perturbation of the global methane burden, showing how the contribution of emissions from unmitigated sectors would respond to mitigation of methane emissions.

The tagging approach and model setup is described in Section 2. In Section 3 we evaluate our simulations against observations from TOAR and the ensemble of simulations from HTAP Phase 2, show the intercontinental source attribution for ozone and its precursors, and examine the response of this source attribution to a 20% perturbation in the global methane burden.

Conclusions are drawn in Section 4.

## 2  Experiment design

Simulations are performed with CAM4-chem (Community Atmosphere Model version 4 with chemistry), a component of the CESM (the Community Earth System Model) version 1.2.2 (Tilmes et al., 2015; Lamarque et al., 2012) using the same model configuration as Butler et al. (2018). The model is run at a horizontal resolution of $1.9 \times 2.5$ degrees, with 56 vertical levels

using specified dynamics for the year 2010 from the MERRA reanalysis (Rienecker et al., 2011). As in Butler et al. (2018), we have replaced the default chemical mechanism with a tagged mechanism based on an earlier version of the MOZART-4 mechanism Emmons et al. (2012). Our tagging system allows the attribution of tropospheric ozone to chemical production by either $NO_x$ or reactive carbon precursors (as well as transport from the stratosphere). A complete attribution of tropospheric ozone to both kinds of precursors requires two model runs: one with $NO_x$ emissions tagged; and another with reactive carbon

emissions tagged. Chemical production of ozone in the stratosphere (primarily through photolysis of molecular oxygen), and

other minor production pathways for tropospheric ozone are also tagged, as described in Butler et al. (2018). For both $NO_x$ tagging and VOC tagging, the sum of the tagged ozone tracers is equal to the total ozone as simulated by the model.

As in Butler et al. (2018), anthropogenic emissions of $NO_x$, CO, and NMVOC for 2010 are taken from the EDGAR-HTAPv2 emission inventory (Janssens-Maenhout et al., 2015), biomass burning emissions are from GFEDv3 (van der Werf et al., 2010), and methane is held fixed at the surface to a global average value of 1760 ppb, as in Tilmes et al. (2015). Two simulations (base runs) are performed with this model setup: one in which all sources of $NO_x$ are tagged as described below (the "$NO_x$-tagged" run); and one in which all sources of reactive carbon are tagged as described below (the "VOC-tagged" run). As in Butler et al. (2018), the length of the spinup period was one year for the $NO_x$-tagged run, and two years for the VOC-tagged run. The model was deemed to be spun up when the maximum difference between the simulated December mean surface ozone attributable to any tagged source was less than 1% in any two subsequent years of simulation.

With the exception of surface-based anthropogenic emissions of $NO_x$, CO, and NMVOC, the tag identities used in this study are identical to those used in Butler et al. (2018). In this study, all surface-based anthropogenic emissions are tagged with a label representing the geographical location at which the emissions occur. This approach allows attribution of simulated ozone to anthropogenic precursor emissions from specific locations. Specifically, anthropogenic emissions of $NO_x$ and reactive carbon are tagged according to their Tier 1 Source Region as defined for the HTAP phase 2 multi-model ensemble experiment, which is described in more detail in (Galmarini et al., 2017). Due to computational constraints, not all of the HTAP Tier 1 regions are tagged in this study. Since the primary focus of this study is on the attribution of ground-level ozone in the Northern Hemisphere, only the major anthropogenic Northern Hemisphere source regions are tagged, while other anthropogenic sources are tagged with the label "Rest of the World". A full list of the tags used in the $NO_x$- and VOC-tagged runs is given in Table 1. The explicitly tagged source regions differ between the $NO_x$-tagged and VOC-tagged runs because VOC tagging is computationally more expensive than the $NO_x$-tagging (Butler et al., 2018). One important difference between this study and Butler et al. (2018) is that anthropogenic emissions of CO for each source region are tagged together with emissions of NMVOC in this study in order to save computational resources. For the emissions tagged as "Oceanic emissions" (Table 1) we note that the only source of $NO_x$ from this region in our simulations is from shipping, and that the major source of reactive carbon is biogenic emissions of dimethyl sulphide (DMS).

In addition to the $NO_x$- and VOC-tagged base runs described above, we also perform two additional runs in order to investigate the response of tropospheric ozone to a perturbation in the tropospheric burden of methane: one with $NO_x$ tagging; and another with VOC tagging. In each of these methane perturbation runs, the initial atmospheric methane burden and the methane mixing ratio imposed at the surface as a boundary condition are reduced by 20%. This translates to a surface methane mixing ratio of 1410 ppb in these methane perturbation runs. In these methane perturbation runs, all other sources of $NO_x$ and reactive carbon are left unchanged. The methane perturbation runs also require two years of spinup for the model to arrive at steady state.

CAM4-chem in version 1.2.2 of the CESM has previously been evaluated by Tilmes et al. (2015), and the modified version used in this study has also been discussed thoroughly by Butler et al. (2018). In Section 3, we describe the key differences in methane and tropospheric ozone between our base simulation and the CAM4-chem simulation reported by Tilmes et al. (2015),

and compare our simulated surface ozone with observations from TOAR (Schultz et al., 2017) as well as with the ensemble of CTM simulations from the HTAP phase 2 multi-model study (Galmarini et al., 2017). The full set of CTMs participating in the HTAP phase 2 multi-model ensemble is given in Table 3 of Galmarini et al. (2017). In this study we compare surface ozone from our base simulation with results from a subset of twelve CTMs: CAM-chem (simulations performed by NCAR); CHASER_re1; CHASER_t106; C-IFS; C-IFS_v2; EMEP_rv4.5; EMEP_rv48; GEMMACH; GEOS-Chem-ADJOINT; GEOS-Chem; OsloCTM3.v2; RAQMS. Details of the configurations used by each of these models in the HTAP phase 2 ensemble can be found in Galmarini et al. (2017), and references therein.

## 3 Results and discussion

All results presented in this study are based on the definition of the troposphere as the model grid cells below the level of 150 ppb of ozone. By design, the ozone simulated in our base model runs is identical with the simulation reported in Butler et al. (2018). Our simulation for 2010 produces a tropospheric ozone burden of 319 Tg($O_3$), which is within one standard deviation of the multi-model mean reported by Young et al. (2013) for the year 2000 ($337 \pm 23$ Tg($O_3$)). Our simulated tropospheric ozone burden is slightly higher than the burden reported by Tilmes et al. (2015) using a similar model setup (309 Tg($O_3$)), which could be due to the use of different emissions datasets. Our simulated tropospheric methane burden (4150 Tg($CH_4$) is the same as reported by Tilmes et al. (2015), but our methane lifetime (due to oxidation in the troposphere by OH), at 7.59 years, is shorter than the 8.82 years reported by Tilmes et al. (2015), likely also due to the use of different emission datasets. Our methane lifetime is towards the lower end of the range (7.1 – 10.6 years) simulated in CTMs, as reported by Saunois et al. (2016).

In Figure 1 we compare our simulated monthly mean surface ozone mixing ratio for 2010 with data from TOAR and with the other models in the HTAP phase 2 CTM ensemble. Results are shown averaged over HTAP Tier 2 receptor regions, and only include grid cells for which TOAR observations are available. In general, most of the HTAP models overestimate the monthly mean surface ozone mixing ratio in regions for which observations are available, consistent with the high model bias reported by Young et al. (2018). Also apparent from Figure 1 is the large range in simulated surface ozone between members of the HTAP model ensemble, which is especially high in the northern spring, approaching a spread of approximately 30 ppb between the lowest and highest ensemble members, which is of a similar order to the northern hemisphere annual mean surface ozone itself. Our modelled monthly average surface ozone mixing ratio in the HTAP Tier 2 receptor regions is generally close to the HTAP ensemble mean, and usually within one standard deviation of the ensemble mean.

### 3.1 Source attribution of tropospheric ozone

The attribution of annual average tropospheric ozone to emissions of $NO_x$ and reactive carbon precursors based on the source tags from Table 1 is shown in Figures 2 and 3 and quantified in Tables 2 and 3. Figures showing the attribution of monthly mean ozone are available in the Supplementary Material. For each tagged source, Tables 2 and 3 include the emissions of $NO_x$ and reactive carbon (respectively, and where applicable), the contribution of each source to the 2010 average tropospheric

ozone burden, the contribution to the Northern Hemisphere 2010 annual average surface mixing ratio, and (where applicable) the ozone production efficiency of each emission source (defined here as the contribution of each emission source to the tropospheric ozone burden, with units of moles of ozone per mole of N or C emitted). Figures 2 and 3 show the spatial distribution of the annual average surface ozone as attributed to each source of $NO_x$ and reactive carbon, respectively. In Figure 2, ozone attributable to anthropogenic $NO_x$ emissions in some source regions (specifically South East Asia, Northern Africa, the Middle East, Middle America, and Central Asia) has been added to the "Rest of the world" total in order to unify the definition of this source region with the definition of this region in the VOC-tagged run. The difference in the stratospheric contribution between the $NO_x$- and VOC-tagged runs is due to the role of $NO_x$ produced in the stratosphere from dissociation of $N_2O$. Ozone produced in reactions involving this stratospheric source of $NO_x$ are counted in our source attribution as stratospheric ozone, as described in Butler et al. (2018).

### 3.1.1 Attribution to $NO_x$ emissions

Anthropogenic $NO_x$ emissions from the three major high-latitude source regions (Europe, East Asia, and North America) contribute to high modelled ozone concentrations both locally and in the Northern Hemisphere background. Lightning $NO_x$, soil $NO_x$, and ozone input from the stratosphere all contribute additionally to modelled global background ozone. Emissions of $NO_x$ from shipping contribute significantly to ozone over the major northern hemisphere ocean basins, which is also transported over continental regions. South Asia stands out in comparison with the other major Northern Hemisphere source regions, in that ozone produced from $NO_x$ emitted in South Asia is relatively localised to the South Asian region itself, and not transported into the hemispheric background to the same extent as ozone produced from $NO_x$ emissions in the other major Northern Hemisphere source regions.

Table 2 shows that $NO_x$ emissions from lightning and aircraft are especially efficient at producing ozone in the free troposphere, consistent with previous work (eg. Beck et al., 1992; Jacob et al., 1996; Dahlmann et al., 2011). Similarly, surface emissions of $NO_x$ from regions closer to the tropics (eg. South East Asia and Middle America) produce ozone more effectively due to rapid convective transport of emitted $NO_x$ into the free troposphere, consistent with Zhang et al. (2016). Of the major Northern Hemisphere source regions, $NO_x$ emissions from South Asia are the most efficient at producing ozone, consistent with a stronger role of vertical transport over this region. In contrast, $NO_x$ emissions from the major anthropogenic source regions in the high northern latitudes (Europe, East Asia, and North America) are among the least productive of all global $NO_x$ emissions, consistent with a relatively small amount of convective transport, leading to higher rates of $NO_x$ removal. Despite their low ozone production efficiency, emissions of $NO_x$ in the high northern latitudes contribute significantly to surface ozone across the northern hemisphere (Figure 2 and Table 2).

Table 2 also shows that $NO_x$ emissions from shipping are also relatively efficient at producing ozone, which is also consistent with previous work (eg. Lawrence and Crutzen, 1999; Hoor et al., 2009). The high ozone production efficiency of ship emissions is due to their location in relatively pristine regions with few other sources of $NO_x$. Due to the high ozone productivity of ship emissions, and being emitted at relatively high latitudes, they contribute significantly to the Northern Hemispheric

background (Figure 2 and Table 2). As noted above, the ozone production from ship $NO_x$ is likely to be overestimated due to the artificial dilution of emissions into relatively coarse model grid cells.

Mertens et al. (2018) report a contribution of shipping to the tropospheric ozone burden of 18 $Tg(O_3)$ using their tagging technique, and based on a model simulation with ship $NO_x$ emissions of 6 $Tg(N)yr^{-1}$. In our study, we calculate a contribution of ship $NO_x$ to tropospheric ozone of 19.9 $Tg(O_3)$ based on ship $NO_x$ emissions of 4.28 $Tg(N)yr^{-1}$, implying a much higher ozone production efficiency for ship $NO_x$ in our study. Since the tagging technique used by Mertens et al. (2018) is based on the technique described by Grewe et al. (2017), which combines the effects of tagged $NO_x$ and reactive carbon precursors into a single tagged ozone molecule during ozone production, we do not expect our results to be directly comparable. Since shipping emits significantly more $NO_x$ than reactive carbon, we would expect the combinatorial tagging approach of Mertens et al. (2018) to attribute less ozone to shipping than our method, as the ozone produced from ship $NO_x$ would also be partially attributed to the reactive carbon precursor involved in the ozone production. Indeed, Mertens et al. (2018) report maximum contributions of shipping to surface ozone of about 10 ppb in summer over major Northern Hemisphere ocean basins. In our study, surface ozone attributable to shipping over these regions can exceed 20 ppb (see the Supplementary Material).

### 3.1.2 Attribution to reactive carbon emissions

Methane and biogenic emissions clearly stand out as major reactive carbon precursors to tropospheric ozone, contributing 35% and 24% respectively to the tropospheric ozone burden in our simulation. Anthropogenic emissions of reactive carbon (excluding biomass burning) together contribute about 14 % to the tropospheric ozone burden. The relatively low influence of anthropogenic reactive carbon emissions on ground-level ozone has been noted elsewhere (eg. HTAP, 2010; Butler et al., 2018), but despite this low overall ozone productivity, anthropogenic reactive carbon emissions from source regions in higher northern latitudes still contribute disproportionately highly to surface ozone in the Northern Hemisphere (Table 3).

Due to the emissions of CO being tagged together with emitted VOC in this study, the contribution of each tagged source to the tropospheric ozone burden (and therefore also the ozone production efficiency of each tagged source) is a mixture of ozone production due to emitted CO and emitted NMVOC. The ozone attributed to methane oxidation in Table 3 is due do all stages of methane oxidation in the MOZART-4 chemical mechanism, including the final step in which CO from earlier stages of methane oxidation is itself oxidised to $CO_2$. The oxidation of CO can produce at maximum one peroxy radical ($HO_2$). The maximum ozone production potential of CO is therefore 1 mole of ozone per mole of emitted CO. VOC (including methane) can produce significantly more ozone per mole emitted carbon, when taking into account the subsequent oxidation of the initial oxidation products (Bowman and Seinfeld, 1994; Atkinson, 2000; Butler et al., 2011). Future studies using this tagging methodology should consider tagging CO emissions separately from NMVOC emissions if they aim to determine the ozone production efficiency of anthropogenic NMVOC emissions from different world regions. Butler et al. (2018) did tag NMVOC emissions separately from CO emissions, but did not tag anthropogenic emissions separately according to their geographical region. We reexamined the output of the otherwise identical VOC-tagged run described by Butler et al. (2018) in order to determine the ozone production efficiency of NMVOC emissions from anthropogenic, biomass burning, and biogenic sources. Respectively, these are 0.0580, 0.0354, and 0.0268 $(mol(O_3)/mol(C))$. The ozone production efficiency of biogenic NMVOC recalculated

from Butler et al. (2018) is not significantly different from the value reported here in Table 3, reflecting the relative minor contribution of CO to the total amount of emitted biogenic reactive carbon. For biomass burning and anthropogenic sources however, the ozone production efficiency of NMVOC emitted from these sources is greater than the corresponding value from Table 3, reflecting the fact that the numbers from Table 3 also include emissions of CO.

We note that methane has a higher ozone production efficiency per unit of reactive carbon (0.0689 mol/mol(C), Table 3) than any of the NMVOC in our runs. The low ozone production efficiency of biogenic NMVOC is consistent with large amounts of isoprene being emitted in remote regions under low-$NO_x$ conditions, where loss of peroxy radicals through reaction with other peroxy radicals could be expected to dominate (Atkinson, 2000). It might however be expected that anthropogenic NMVOC would have a higher ozone production efficiency, due to their being co-emitted with anthropogenic $NO_x$, favouring the conversion of NO to $NO_2$ through reaction with peroxy radicals, and thus the production of ozone. The relatively low production efficiency of anthropogenic NMVOC in our model runs could be due to the relatively simple chemistry of methane oxidation being well-described in the version of the MOZART-4 chemical mechanism used here, in which the relatively complex chemistry of the higher NMVOC has been simplified. Coates and Butler (2015) noted that the ozone production potential of NMVOC in simplified chemical mechanisms tended to be lower than the more comprehensive Master Chemical Mechanism (Saunders et al., 2003). Utembe et al. (2010) previously noted increased tropospheric ozone in a CTM when using a more explicit oxidation mechanism for NMVOC. The extremely low ozone production efficiency of reactive carbon from oceanic sources in Table 3 is due to the lack of any ozone forming pathways in the oxidation of dimethyl sulphide (DMS) in the MOZART-4 chemical mechanism as used in this study. DMS is the dominant source of reactive carbon over the oceans in our model simulations.

To our knowledge, the only other study to perform source attribution of global tropospheric ozone specifically to reactive carbon precursors is Butler et al. (2018), on which the present study builds. Here, we attribute 113 Tg of ozone to methane oxidation (Table 3). Grewe et al. (2017) attribute 45 Tg of ozone to methane using their tagging approach, which combines the effects of tagged $NO_x$ and reactive carbon precursors into a single tagged ozone molecule during ozone production. Ozone production due to methane oxidation under their combinatorial tagging approach would be expected to also include attribution to the source of $NO_x$ involved in the ozone production. We would thus expect Grewe et al. (2017) to attribute approximately half of the amount of ozone to methane as we would. Doubling the value reported by Grewe et al. (2017) yields 90 Tg of ozone attributable to methane oxidation, which is much closer to our value of 113 Tg.

Widespread implementation of tagging techniques for separated tagging of $NO_x$ and reactive carbon emissions in other CTMs, along with systematic inter-comparisons of their results could help to understand differences in the simulated budgets of tropospheric ozone. Due to the variety of approaches taken by different tagging techniques (summarised in Butler et al., 2018), an inter-comparison of a range of different tagging techniques with other methods for ozone source attribution could be informative.

### 3.2 Source-receptor relationships for ozone

#### 3.2.1 Annual average surface ozone

Figure 4 shows the modelled annual average surface ozone concentration in the five major HTAP Tier 1 regions in the northern hemisphere (Europe, Russia/Belarus/Ukraine, South Asia, East Asia, and North America), including a full attribution of ozone in each of these regions to all sources, including transport from the stratosphere and emitted precursors of both $NO_x$ and reactive carbon. Annual average ozone in most of the regions shown in Figure 4 is close to the Northern Hemisphere annual average of 30 ppb (Table 2), except in South Asia and East Asia, where the annual average surface ozone mixing ratio is closer to 40 ppb. The difference in each case is primarily due to a larger source of ozone produced from locally emitted precursors. Transport from the stratosphere contributes approximately 2–4 ppb to annual average surface ozone depending on the receptor region, consistent with the 2.91 ppb contribution of stratospheric ozone to the annual average surface ozone in the Northern Hemisphere average surface ozone (Table 3). As already shown in the previous section, anthropogenic sources of $NO_x$ dominate other $NO_x$ sources as ozone precursors, while the major reactive carbon precursors are methane and BVOC.

In each of the five regions shown in Figure 4, natural sources and long-range transport of ozone produced from extra-regional anthropogenic precursors together contribute more to the annual average surface ozone than anthropogenic emissions within the region itself. In each region, the local anthropogenic $NO_x$ emissions produce more ozone than can be attributed to anthropogenic $NO_x$ emissions in any other Tier 1 regions, but with only one exception (South Asia), the combined contribution of external anthropogenic $NO_x$ emissions to annual average surface ozone is greater than the local contribution. The importance of long-range transboundary transport of ozone has been noted elsewhere (HTAP, 2010).

While anthropogenic precursor emissions from South Asia contribute significantly to surface ozone within the South Asia region, they contribute relatively little to surface ozone in the other four regions shown in Figure 4. This is also consistent with the surface ozone maps in Figure 2, and the higher ozone productivity of $NO_x$ emissions from South Asia when compared with the other major Northern Hemisphere source regions (Table 2). Emissions of ozone precursors (particularly $NO_x$) from South Asia are transported efficiently into the free troposphere, where they contribute disproportionately to the global tropospheric ozone burden (as also noted by Zhang et al., 2016), but the contribution of South Asian emissions to surface ozone in other parts of the Northern Hemisphere is disproportionately smaller than emissions from the other HTAP Tier 1 regions.

#### 3.2.2 Seasonal cycles of surface ozone

Figures 5, 6, and 7 show the seasonal cycles of surface ozone in the three selected Tier 2 regions "North West Europe", "North East China", and "North West United States" respectively. These regions are selected in order to compare two regions on the western side of their respective continents (where long-range transport is expected to be important) with a region on the eastern side of its continent that is also a major source region. A set of figures for other HTAP Tier 2 regions, with a complete attribution of surface ozone to all tagged HTAP Tier 1 source regions is available in the Supplementary Material. In each of Figures 5, 6, and 7, results are shown for both $NO_x$- and VOC-tagging. In each receptor region, the contribution of long-range transport due to extra-regional anthropogenic emissions from HTAP Tier 1 regions is shown both in aggregate (top panels) and

360 by individual Tier 1 source region (bottom panels). The definition of the "Rest of the world" ozone tracer has been harmonised between the $NO_x$- and VOC-tagged runs in these figures. Consistent with the annual averages from Figure 4, anthropogenic $NO_x$ sources also dominate the seasonal cycle of modelled ozone, while the major reactive carbon precursors of ozone are methane and BVOC.

All three receptor regions show a seasonal cycle of ozone with a spring-summer ozone maximum superimposed on a year-
365 round ozone baseline. The summertime maximum in ozone is clearly due to local photochemical production from the combination of locally-emitted anthropogenic $NO_x$ and biogenic VOC. The strong role of locally-emitted precursors in the production of ozone in summer is consistent with earlier work (eg. Reidmiller et al., 2009; Huang et al., 2017; Jonson et al., 2018; Han et al., 2019), while the importance of biogenic VOC emissions, especially isoprene, for ozone production in summer has also been noted elsewhere (eg. Chameides et al., 1992; Andersson and Engardt, 2010; Han et al., 2019). Biogenic emissions of $NO_x$
(from soils) also contribute to this summertime maximum in local photochemical ozone production in all three of the regions shown in Figures 5, 6, and 7, but to a much smaller extent than anthropogenic $NO_x$ emissions.

The year-round baseline ozone in our model simulations in all three receptor regions can be primarily explained by slower photochemistry involving methane as the reactive carbon precursor, in combination with extra-regional anthropogenic $NO_x$ (Figures 5, 6, and 7). The contribution of methane to surface ozone is slightly larger in summer, coinciding with the peak
in local anthropogenic $NO_x$ emissions, consistent with local photochemical ozone production from enhanced local methane oxidation. The contribution of extra-regional anthropogenic $NO_x$ to surface ozone is largest in spring, coinciding with the peak in the contribution of extra-regional anthropogenic reactive carbon, consistent with long-range transboundary transport of ozone produced elsewhere.

Maxima in springtime ozone have previously been linked to long-range transboundary transport in all major receptor regions
(HTAP, 2010; Lin et al., 2012; Jonson et al., 2018; Ni et al., 2018). This transported ozone can be attributed to input from the stratosphere, as well as extra-regional anthropogenic emissions of $NO_x$ and reactive carbon. In our simulations, the contribution of stratospheric ozone peaks around March, while the contribution of extra-regional anthropogenic emissions tends to peak around April, when it contributes more strongly to monthly average surface ozone in each region than local anthropogenic $NO_x$ (Figures 5, 6, and 7). In all Northern Hemisphere regions, the springtime peak in the contribution of extra-regional anthro-
pogenic reactive carbon is smaller than the corresponding springtime peak in the contribution of extra-regional anthropogenic $NO_x$. Previous work has identified uncertainties in the treatment of ozone production from NMVOC oxidation as a potential source of inter-model differences (eg. Emmerson and Evans, 2009; Utembe et al., 2010; Coates and Butler, 2015). The relatively large influence of anthropogenic NMVOC on springtime ozone (compared with its influence during other time of the year) could be a contributing factor to the large spread in springtime ozone simulated by current generation CTMs (Figure 1).

$NO_x$ from shipping is the largest single contributor to springtime transboundary ozone transport in all three receptor regions shown here. We note however that the coarse resolution of our model (2 degrees) would be expected to exaggerate the effects of ship $NO_x$ on ozone production due to rapid dilution of the emissions (von Glasow et al., 2003), as well as exaggerate the transport of $NO_x$ and ozone near coastlines due to unrealistically high diffusion between adjacent land and ocean grid cells. The contribution of shipping emissions to surface ozone in our simulations should thus be considered an upper bound, especially

in coastal regions. The high contribution of ship $NO_x$ to summertime ozone in Europe (Figure 5) may be an artefact of coarse model resolution, combined with high shipping volume near coasts in the Eastern North Atlantic Ocean as well as the North Sea. Early work by Lawrence and Crutzen (1999) showed a stronger influence of ship $NO_x$ on surface ozone in North West Europe than on any other continental region. Jonson et al. (2018) show that the only other CTM in the HTAP phase 2 ensemble to report the results of a perturbation of shipping emissions (the EMEP_rv48 CTM with a resolution of $0.5 \times 0.5$ degrees) shows a similar magnitude for the influence of ship $NO_x$ on summertime ozone in Europe as for springtime ozone. Lupaşcu and Butler (2019), using a regional model at $50 \times 50$ km resolution and a similar ozone tagging system as used in the present study, showed that the contribution of ship $NO_x$ to ozone in coastal regions of Europe reaches a maximum level in summer. Jonson et al. (2020), using a global model with a resolution of $0.5 \times 0.5$ degrees showed that shipping near coastal strongly influences ozone over North West Europe in both spring and summer, while $NO_x$ emissions from shipping on the high seas have a stronger influence on European ozone in spring than in summer. In contrast, the study of Aksoyoglu et al. (2016), using a higher-resolution regional model ($20 \times 20$ km) for Europe, does not indicate such a strong role for ship $NO_x$ on summertime ozone over Europe.

In other receptor regions, the influence of ship $NO_x$ emissions on surface ozone is largest in spring (Figures 6 and 7), suggesting a stronger influence of $NO_x$ emissions over the high seas on springtime ozone in our simulations in these regions. Model-dependent inconsistencies in the treatment of ship $NO_x$ emissions may play a role in the large spread of simulated ozone between models in springtime (Figure 1). Future work should examine the contribution of ship $NO_x$ emissions to ozone in both spring and summer, using model systems which include both better representations of plume dilution over the major shipping routes, more refined attribution to shipping emissions from coastal regions and the high seas, as well as higher resolution over receptor regions.

Previous work has indicated a strong influence of anthropogenic emissions from both North America and East Asia on springtime ozone in Europe (Jonson et al., 2018), a strong influence of East Asian emissions on springtime ozone in North America (Lin et al., 2012), and a diverse range of intercontinental influences on springtime ozone in East Asia (Ni et al., 2018). Direct numerical comparison of our results with these previous studies is difficult due to the different methodologies used. These previous studies all employed the perturbation technique to determine the influence of all anthropogenic emissions (both $NO_x$ and reactive carbon) from each source region, while this work uses a tagging approach which separately attributes ozone to emitted $NO_x$ and reactive carbon. Qualitatively, our results as shown in Figures 5, 6, and 7 do however appear consistent with this earlier work. Comparison of the $NO_x$-tagged and VOC-tagged results in these Figures shows that anthropogenic $NO_x$ emissions from most source regions have a stronger influence on springtime ozone in any given receptor region than anthropogenic emissions of reactive carbon. The only exception to this is East Asia, where reactive carbon emissions are substantially higher than in other HTAP Tier 1 source regions (Table 3). Reactive carbon emissions from East Asia contribute approximately equally to springtime ozone in North America as East Asian $NO_x$ emissions, and in Europe, East Asian reactive carbon contributes more to springtime ozone than East Asian $NO_x$.

### 3.3 Long-range transport of ozone precursors

Fiore et al. (2018) has suggested that measurements of the abundance of PAN at mountaintop sites in spring may be useful as an indicator of intercontinental transport of ozone and its precursors, as well as being a diagnostic for uncertainties in CTM simulations, which show large inter-model differences in simulated PAN (Emmons et al., 2015). The column integrated density of PAN in the lower troposphere (defined here as the model layers between 500 and 800 hPa) in the three HTAP Tier 2 receptor regions "North West Europe", "North East China", and "North West United States" are shown in Figures 8, 9, 10. A set of figures for other HTAP Tier 2 regions, with a complete attribution of surface ozone to all tagged HTAP Tier 1 source regions is available in the Supplementary Material. Simulated PAN is highest in late winter to early spring, consistent with earlier work (Fischer et al., 2014; Fiore et al., 2018). The extra-regional contribution to PAN is also highest in spring, and this is due primarily to anthropogenic NMVOC, also consistent with Fischer et al. (2014).

Our model simulations with $NO_x$- and VOC tagging provide a unique opportunity to examine the origin and fate of PAN as simulated in our model, since this allows simultaneous attribution of simulated PAN to both is $NO_x$ precursor and its reactive carbon precursor. Comparison of the bottom two panels in each of Figures 8, 9, 10 shows consistently that for any given land-based HTAP Tier 1 source region, the anthropogenic NMVOC emissions contribute more to PAN formation than the anthropogenic $NO_x$ emissions from that region to the PAN modelled in all HTAP Tier 2 receptor regions. The balance of extra-regional PAN in all cases is due to $NO_x$ emissions from shipping. In our simulations, significant amounts of PAN are formed downwind of the regions in which the anthropogenic NMVOC precursors are emitted, often through reaction with $NO_x$ emitted from shipping. A strong influence of anthropogenic $NO_x$ emissions on PAN in the northern mid-latitudes is consistent with the results shown by Fischer et al. (2014) (their Figure 7, which does not distinguish between different sources of anthropogenic $NO_x$).

Figures 8, 9, and 10 also show that the reactive carbon component of PAN is generally more persistent than the $NO_x$ component. For example, the contribution of anthropogenic NMVOC from North America to springtime PAN over East Asia is only slightly lower than its contribution to springtime PAN over Europe, which is much closer to North America considering the prevailing westerly winds (bottom right panels of Figures 8 and 9). In contrast, the contribution of anthropogenic $NO_x$ from North America to springtime PAN in East Asia is substantially less than its contribution to springtime PAN over Europe (bottom left panels of Figures 8 and 9).

### 3.4 Attribution of Northern Hemisphere total organic reactivity

We examine the Northern Hemisphere budget of reactive carbon in more detail in Figure 11. This figure shows the seasonal cycle of the Northern Hemisphere column-integrated total reactivity with respect to the OH radical of all reactive carbon containing species in our simulation, attributed to their emission source. The total OH reactivity of reactive carbon species of an airmass is often linked to its ozone production potential (Chameides et al., 1992; Kleinman et al., 2002). The OH reactivities shown in Figure 11 include in each case the OH reactivity of the primary emitted species, as well as the OH reactivity of each carbon-containing oxidation product. These were calculated using monthly averaged output of the modelled concentration of

each carbon-containing species (including its associated tags), and the temperature- and pressure-dependent rate coefficients for their reaction with the OH radical, then averaged over all Northern Hemisphere grid cells, weighted by air density.

The total Northern Hemisphere OH reactivity of reactive carbon remains fairly constant year-round at about $0.6 - 0.7\,\mathrm{s}^{-1}$, but the seasonal cycles of the OH reactivity attributable to different reactive carbon sources show more variability. Methane (and its oxidation products) contribute about $0.2 - 0.3\,\mathrm{s}^{-1}$ (almost half of the total hemispheric reactivity), with a slight maximum in the summer, consistent with enhanced oxidation (and thus enhanced availability of more reactive methane oxidation products) due to higher OH in summer. The contributions of anthropogenic and biogenic reactive carbon sources to total hemispheric reactivity are similar, ranging between about $0.1 - 0.3\,\mathrm{s}^{-1}$, but with distinct seasonal cycles. The reactivity of biogenic carbon is highest in summer-autumn (consistent with the Northern Hemisphere growing season), while reactivity of anthropogenic carbon is highest in winter-spring (consistent with constant year-round anthropogenic emissions, and a build-up of reactive carbon over winter due to lower hemispheric OH). The build-up of anthropogenic reactive carbon throughout the Northern Hemisphere over winter, combined with the resumption of OH chemistry in spring is consistent with the disproportionate effect of extra-regional anthropogenic reactive carbon on springtime ozone seen in Figures 5, 6, and 7. Uncertainties in the model chemical mechanisms associated with the oxidation of anthropogenic NMVOC (eg. Emmerson and Evans, 2009; Utembe et al., 2010; Coates and Butler, 2015) may thus also contribute to the large spread in simulated ozone seen in the HTAP ensemble during spring (Figure 1).

Figure 11 also shows the geographical origin of Northern Hemisphere anthropogenic carbon reactivity. Emissions of reactive carbon from East Asia stand out as the single major source of enhanced anthropogenic carbon reactivity in winter and spring in our simulations. This is consistent with the high emissions of reactive carbon from this region in 2010 noted earlier (Table 3). Growth in NMVOC emissions from East Asia may have continued since this time (Li et al., 2019), while $NO_x$ emissions have been decreasing (Liu et al., 2017). Increasing trends in local production of ozone during summer over East Asia (eg. Li et al., 2018) should be associated with increased oxidation of reactive carbon, and thus potentially less export of reactive carbon into the Northern Hemisphere background during summer. We expect, however, that increasing emissions of reactive carbon in East Asia should lead to an increased build-up of East Asian reactive carbon in the Northern Hemisphere over winter, and thus also to increased East Asian contribution to extra-regional springtime ozone in other parts of the Northern Hemisphere.

Our tagging technique is currently the only one we know of which is capable of examining the budget of reactive carbon in the level of detail presented in this study. The separate tracking of the carbon-containing and nitrogen-containing components of PAN is particularly informative, suggesting that significant amounts of PAN are formed downwind of source regions in our model, especially during winter and spring, due to a build-up of anthropogenic reactive carbon over winter when photochemistry is relatively slow. Given the large variety in model representations of NMVOC chemistry, including PAN formation and decomposition processes (Emmerson and Evans, 2009; Knote et al., 2015) and the large inter-model differences in simulated PAN (Emmons et al., 2015), the widespread implementation of similar tagging diagnostics in other CTMs may help to provide additional information about the origin and fate of simulated PAN, and more generally about the influence of reactive carbon on atmospheric composition. In combination with routine mountaintop observations of springtime PAN, this may aid understanding of the global PAN budget (Fiore et al., 2018) and other processes responsible for intercontinental transport of

air pollution. Better constraints on these chemical and transport processes should also help to reduce inter-model differences in simulated springtime ozone (Figure 1).

### 3.5 Tropospheric ozone sensitivity to methane

We performed an additional set of model runs with both $NO_x$- and VOC-tagging with the methane surface boundary condition reduced from 1760 ppb to 1410 ppb, a reduction of 350 ppb, or 20%. This perturbation can also be expressed as an increase of 25%. Here we interpret the methane perturbation run in terms of the atmospheric response to a 25% increase in the methane surface mixing ratio at steady state.

In response to the 25% increase in the imposed surface mixing ratio of methane, the total tropospheric burden increased by 776 Tg($CH_4$), an increase of 23%. The strength of the annual tropospheric chemical sink of methane due to OH increased by 72.5 Tg($CH_4$), or 15.2%. The corresponding increase in the methane lifetime was 0.48 years, or 6.75%. The relatively small growth in the chemical methane sink compared with the magnitude of the perturbation in methane itself is consistent with the feedback of methane on its own lifetime due to depletion of OH (Prather, 1996). Table 4 shows the response of the tropospheric ozone burden (and the contributions of different reactive carbon precursors) to the 25% increase in the imposed surface mixing ratio of methane. The 1 ppb simulated increase in Northern Hemisphere surface ozone in response to a 25% increase in methane burden is consistent with previous work (HTAP, 2010). The 9.22 Tg increase in tropospheric ozone burden is also consistent with the review of Fiore et al. (2008), who derived a sensitivity of 0.11–0.16 Tg($O_3$) per Tg($CH_4$)yr$^{-1}$ emitted based on an analysis of the literature. We calculate 0.13 Tg($O_3$) per Tg($CH_4$)yr$^{-1}$ based on our results.

The relative increase in tropospheric ozone attributed by our tagging scheme to methane (13.0%) is comparable to, but slightly smaller than the increase in the magnitude of the chemical methane sink due to OH (15.2%), consistent with the troposphere as a whole becoming slightly more $NO_x$-limited with increasing methane. The absolute increase in the total ozone burden (9.22 Tg($O_3$)) is, however, significantly lower than the increase in the burden of ozone attributed by our tagging scheme to methane (13.0 Tg($O_3$)). When the methane burden is increased, the contribution of every other reactive carbon source to the tropospheric ozone burden decreases (each by approximately 1 – 2%) to partially offset the increased ozone production from methane oxidation. This is also consistent with a slightly more $NO_x$-limited atmosphere with increasing methane. In a future with an increased methane burden, control of NMVOC emissions could be expected to be less effective at large-scale reduction in annual average ground-level ozone.

Table 5 shows the change in the contributions of different $NO_x$ sources to tropospheric ozone in response to the 25% increase in methane burden. As expected, all $NO_x$ sources become more productive when the total atmospheric burden of reactive carbon is increased (consistent with the troposphere as a whole becoming more $NO_x$-limited). The increase in the productivity of the different $NO_x$ sources under an increased burden of methane is however not uniform. Ozone production due to $NO_x$ from shipping stands out as highly sensitive to the global methane burden in our simulations. Ship $NO_x$ accounts for almost 30% of the 1 ppb increase in Northern Hemisphere average surface ozone when the methane burden is increased by 25% (Table 5), despite being a much smaller percentage of total global $NO_x$ emissions (Table 2).

The spatial distribution of the increase in annual average surface ozone from ship $NO_x$ in response to the 25% increase in methane is similar to the spatial distribution of surface ozone due to ship $NO_x$ in our base run (Figure 2). Figures showing the response of attributed surface ozone are available in the Supplementary Material. The response is largest over the major Northern Hemisphere ocean basins, but also extends over continental regions. The seasonal cycle of the increase in annual average surface ozone from ship $NO_x$ in the three HTAP Tier 2 regions examined here in response to the 25% increase in methane is similar to the seasonal cycle of surface ozone due to ship $NO_x$ in our base run (Figures 5, 6, and 7). The maximum response of surface ozone from ship $NO_x$ to rising methane is simulated over the major Northern Hemisphere ocean basins in summer (which in our simulations influences surface ozone in North West Europe, Figure 5), while the influence of this response over most Northern Hemisphere continental regions is generally higher in winter-spring (as seen in North East China, Figure 6).

Previous work (Lawrence and Crutzen, 1999) has noted the disproportionate influence of ship $NO_x$ on tropospheric ozone due to the diffuse and widespread nature of this source over regions which would otherwise have very low mixing ratios of $NO_x$. Fiore et al. (2008) noted that the response of surface ozone to increased methane was especially strong in ship tracks. Myhre et al. (2011) also showed that ship $NO_x$ emissions reduce the global methane lifetime much more than terrestrial $NO_x$ emissions. We note again that the contribution of ship $NO_x$ to ozone in our simulations (as in most current-generation CTMs) is likely to be an overestimate due to the unrealistic dilution of these emissions into coarse model grid cells (von Glasow et al., 2003), and the lack of explicit plume chemistry (Vinken et al., 2011). We do expect however, that the interaction between ship $NO_x$ and methane for ozone production would persist in our model even with a more realistic treatment of ship emissions, since this interaction is likely due to the location, rather than the magnitude of ship emissions. We are not aware of any previous work linking the combined influence of these two sources to a potentially disproportionate influence on background ozone in the Northern Hemisphere, and on modelled surface ozone air quality in inhabited regions of the Northern Hemisphere, especially in spring. Given the current uncertainty in attribution of recent trends in methane (Turner et al., 2019) and the potential for future increases in methane emissions, combined with slower reductions in $NO_x$ emissions from international shipping than from other sectors (eg. the SSP5 future emission scenario Rao et al., 2017), we expect that model simulations of future background ozone in the Northern Hemisphere, especially during spring, may come to be increasingly influenced by ozone produced through the interaction of methane and ship $NO_x$. Future work should investigate the ozone production through interaction of these two sources in more detail.

## 4  Conclusions

We have performed a source attribution for tropospheric ozone in a chemical transport model using a novel technique which separately accounts for the influence of both the emitted $NO_x$ and the emitted reactive carbon precursors on simulated tropospheric ozone. By tagging anthropogenic emissions of $NO_x$ and reactive carbon according to their geographical region we have calculated source/receptor relationships for the Northern Hemisphere. The results of our study are consistent with previ-

ous work, and provide a number of important new insights of relevance to both the mitigation of intercontinental transboundary air pollution and ongoing efforts to reduce the uncertainty in the current generation of chemical transport models.

Consistent with previous work, annual average ground-level ozone in all major Northern Hemisphere regions is primarily influenced by extra-regional emissions of both $NO_x$ and reactive carbon. In all cases, local anthropogenic emissions of ozone precursors have a smaller influence on annual average ozone than the combined effect of precursor emissions from the rest of the world. As a reactive carbon precursor, methane contributes 35% of the tropospheric ozone burden, and 41% of the Northern Hemisphere annual average surface mixing ratio, more than any other source of reactive carbon. Our novel tagging methodology also reproduces the well-known dependence of summer ozone maxima on local emissions of anthropogenic $NO_x$ and biogenic reactive carbon, and the enhanced importance of intercontinental transport of ozone from remote anthropogenic sources in spring. Consistent with previous work, we find that emissions of $NO_x$ at low latitudes produce free-tropospheric ozone more effectively due to more efficient vertical transport. We show, however, that $NO_x$ sources at higher northern latitudes have a stronger influence on ground-level ozone, which is known to have a lower radiative forcing but a higher influence human health and ecosystems.

The current generation of chemical transport models has particular difficulty in simulating the intercontinental transport of ozone, as shown by the large spread in ensemble simulations of ground-level ozone during the spring months. We show that our tagging methodology can deliver detailed diagnostic information about the origin and budget of springtime ozone in our model, along with information about the springtime budget of peroxyacetyl nitrate (PAN), which is also associated with springtime long-range transport and ozone production. We show that a substantial proportion of the free-tropospheric PAN simulated by our model in spring is not produced in the polluted boundary layer over the major anthropogenic source regions, but is rather produced in our model downwind of these regions through the interaction of transported anthropogenic reactive carbon and $NO_x$ emitted from international shipping. Reactive carbon of anthropogenic origin (and its oxidation products, including PAN) builds up in our model across the entire Northern Hemisphere during the winter months, and then contributes in our simulations to a short burst of hemispheric-scale ozone production during spring. In all but the most polluted source regions, anthropogenic NMVOC do not make a significant contribution to simulated ground level ozone in any other season but spring.

We showed here that export of anthropogenic reactive carbon from East Asia may be playing a dominant role in contributing to the build up of reactive carbon in the Northern Hemisphere over winter, and thus to the hemispheric-scale production of ground-level ozone in spring. Given the likely lack of recent mitigation in reactive carbon emissions from East Asia, we expect this effect to be ongoing, and recommend that future work continue investigation of this possibility using updated emission inventories.

In addition to a contribution from the stratosphere, the springtime peak in transported ozone in our model is influenced by the interaction of two processes known to be especially poorly represented in current models: the chemistry of the intermediate oxidation products of NMVOC; and the emissions of $NO_x$ from international shipping. Furthermore, the response of ground-level ozone to changes in methane also appears highly sensitive to the treatment of ship $NO_x$, especially in spring. We believe that our tagging technique could deliver useful information about the large differences in simulated springtime ozone between current generation models, if implemented in a larger number of models and used systematically in model inter-comparison

exercises. This could potentially point the way to improved representations of the processes responsible for intercontinental transport of ozone.

Improved global chemical transport models are required to inform effective policies aimed at reducing intercontinental transport of ground-level ozone, a problem which is most urgent in the springtime. In particular, we recommend that developers of emission inventories and CTMs revisit their representations of anthropogenic NMVOC emissions and associated oxidation chemistry in order to reduce the uncertainties in modelled springtime ozone. Additionally, more explicit representations of the $NO_x$ chemistry of ship exhaust plumes should be prioritised in order to improve the suitability of current models for simulating both intercontinental transport of ozone as well as the response of ozone to changing atmospheric methane.

*Code availability.* The CESM is maintained by NCAR, and is provided free to the community. The specific modifications made to the model to enable tagging of ozone production have been described by Butler et al. (2018) and are available in the online supplement to that work, which is archived and accessible through the following DOI: 10.5194/gmd-11-2825-2018

*Author contributions.* TB designed the study. Model runs were performed by AL. Analysis of model runs was performed by AL, AN, and TB. TB wrote the paper with input from AL and AN.

*Competing interests.* The authors declare that they have no competing interests.

*Acknowledgements.* The authors would like to thank Mark Lawrence, Louisa Emmons, Simone Tilmes, and Terry Keating for numerous helpful discussions during the preparation of this manuscript.

*Financial support.* This work was hosted by IASS Potsdam, with financial support provided by the Federal Ministry of Education and Research of Germany (BMBF) and the Ministry for Science, Research and Culture of the State of Brandenburg (MWFK).

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

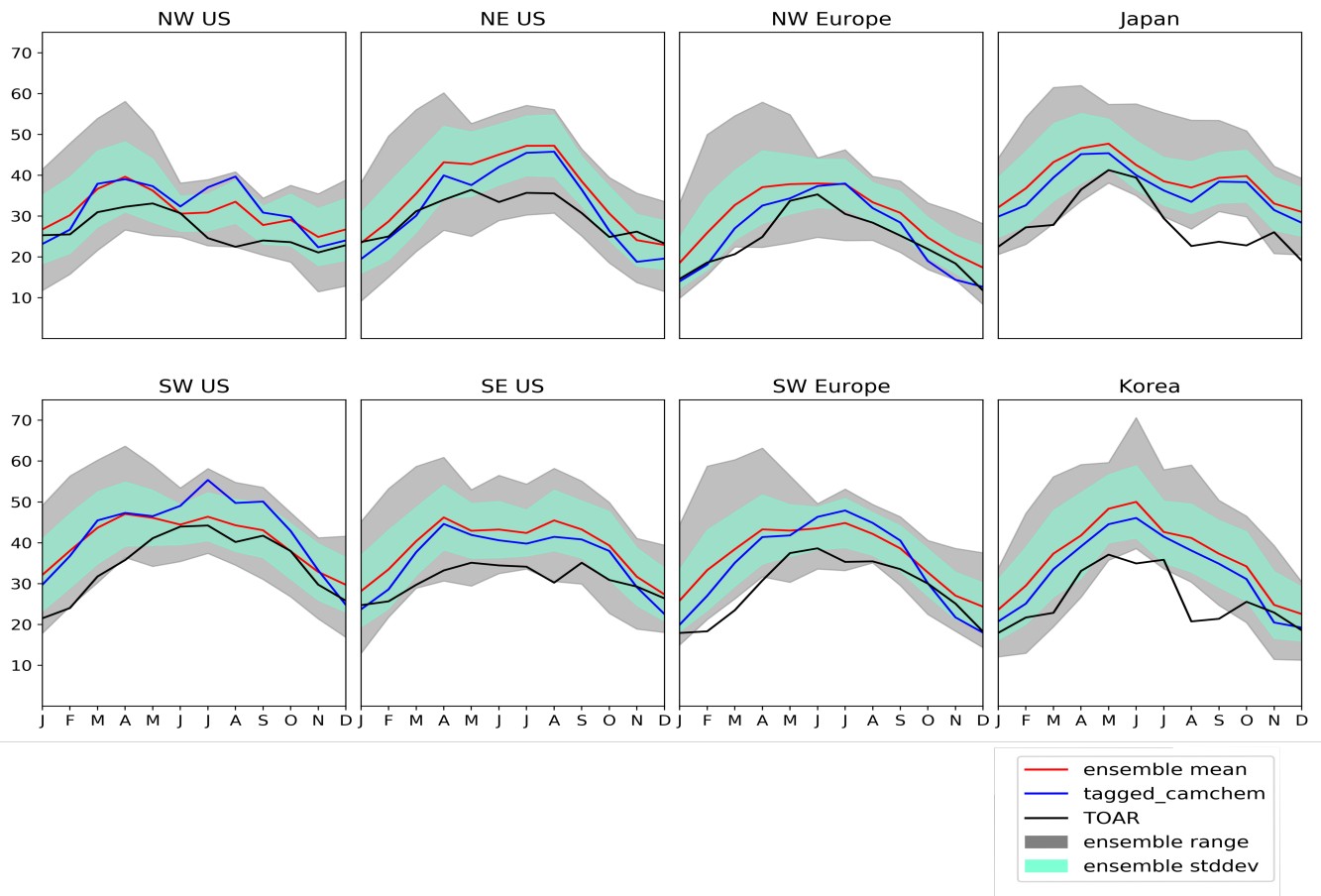

**Figure 1.** Seasonal cycle of monthly mean surface ozone (ppb) in HTAP Tier 2 regions from our base model run (blue line), compared with with observations from TOAR (black line), and other models from the HTAP ensemble of global models: ensemble mean (red line); ensemble standard deviation (green shaded area); and ensemble range (grey shaded area). Only grid cells containing TOAR observations have been used.

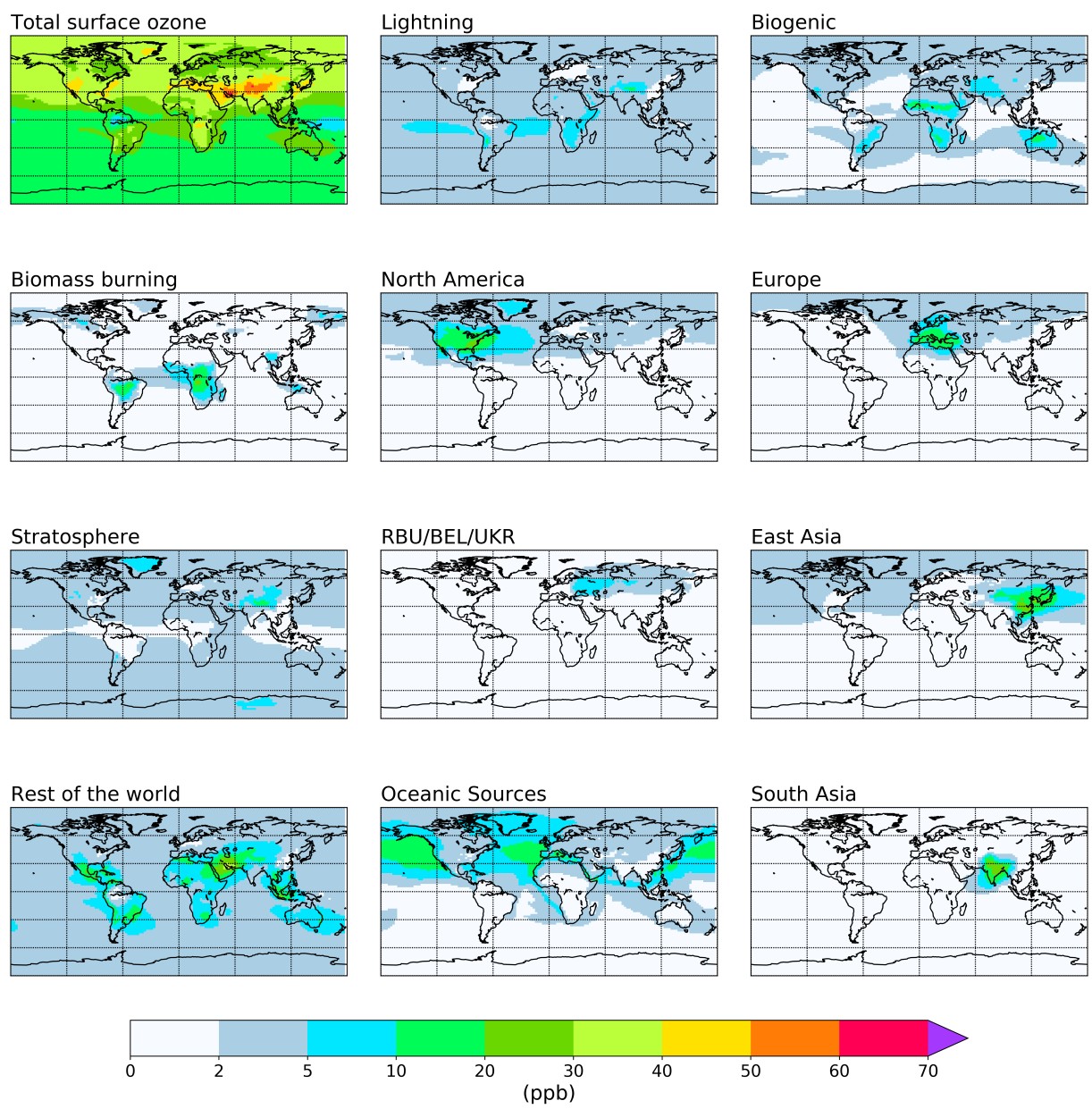

**Figure 2.** Annual mean surface ozone (ppb) from the $NO_x$-tagged base run. Total ozone is shown in the top left panel. Tagged ozone tracers are shown in other panels.

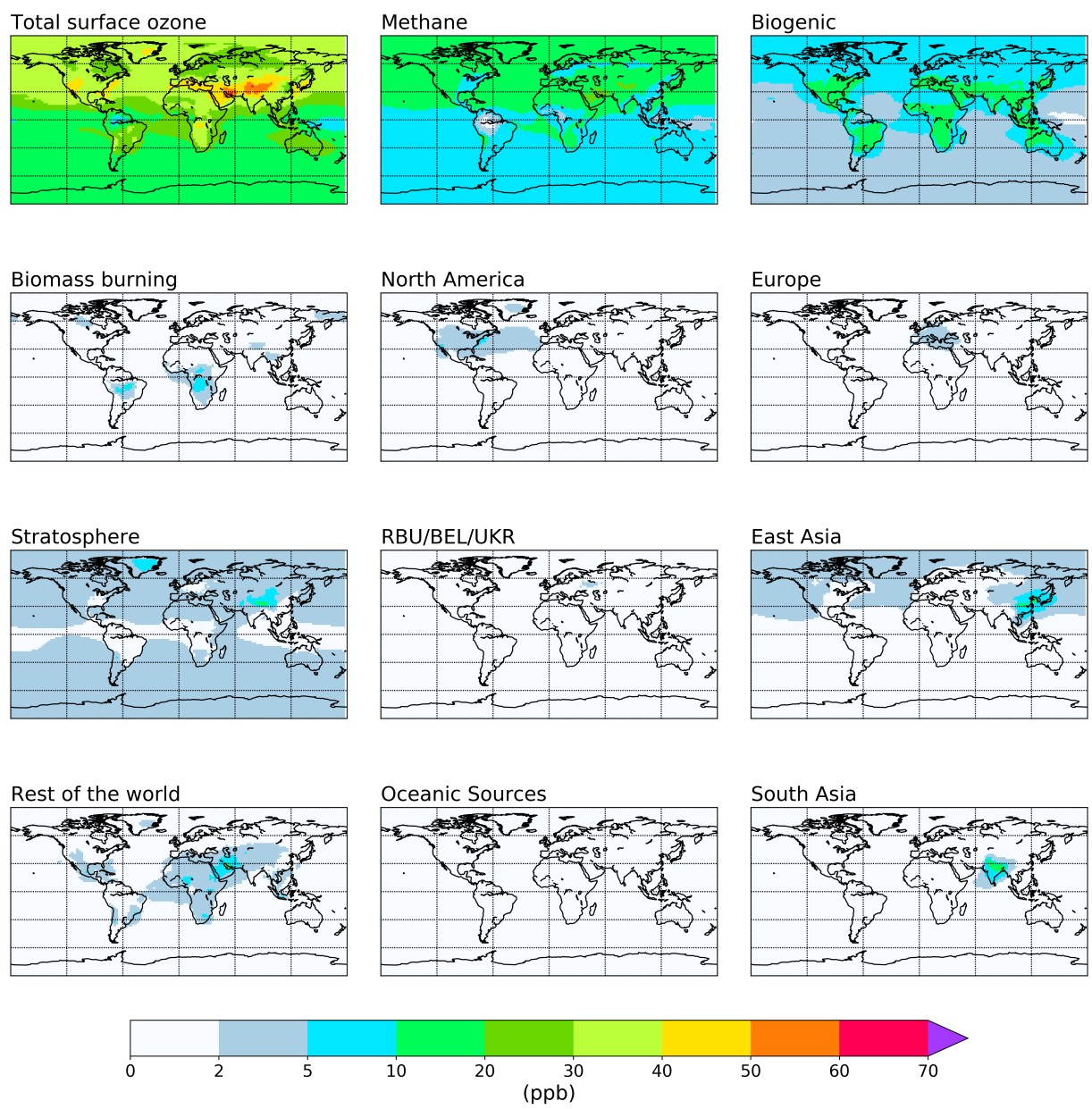

**Figure 3.** Annual mean surface ozone (ppb) from the VOC-tagged base run. Total ozone is shown in the top left panel. Tagged ozone tracers are shown in other panels.

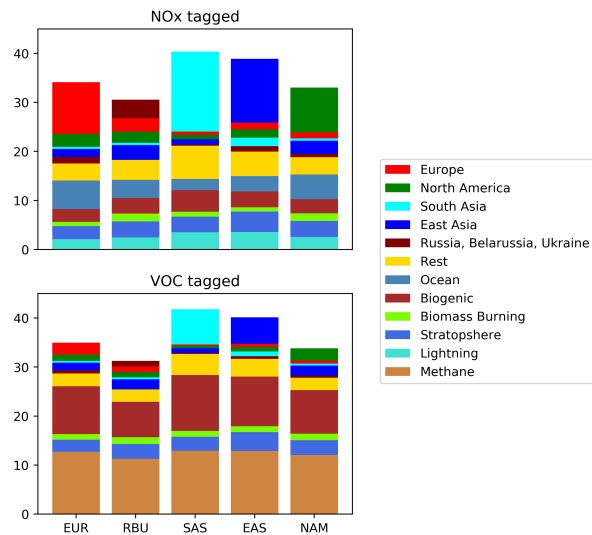

**Figure 4.** Source-receptor relationships for annual average surface ozone (ppb) in major Northern Hemisphere Tier 1 regions EUR (Europe); RBU (Russia, Belarus, and Ukraine); SAS (South Asia); EAS (East Asia); and NAM (North America). The attribution relates the annual average surface ozone modelled in each region to the emitted precursors $NO_x$ (top panel) and reactive carbon (bottom panel) from all HTAP Tier 1 regions.

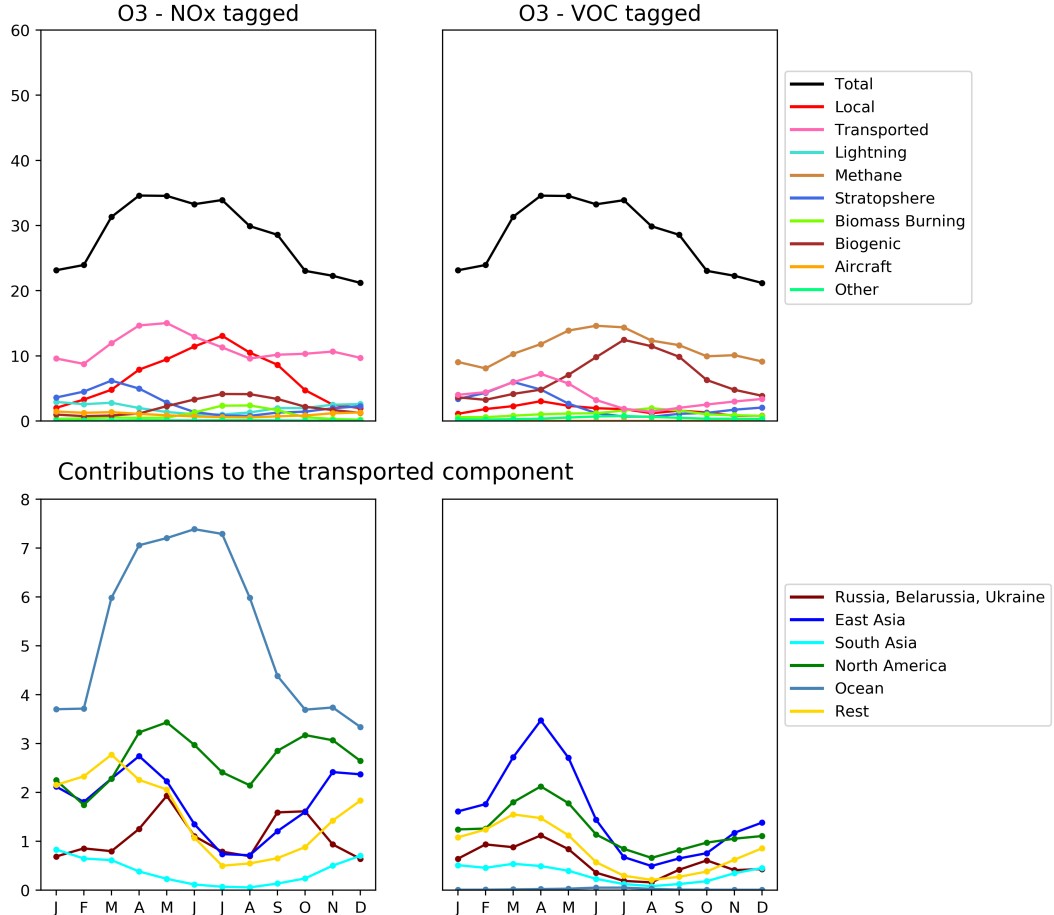

**Figure 5.** Seasonal cycle of surface ozone (ppb) in the HTAP Tier 2 receptor region "North West Europe". $NO_x$-tagging is shown in the left panels, and reactive carbon tagging in the right panels. Top panels show total monthly mean ozone (black line) as well as the local anthropogenic component, long-range transported anthropogenic component, and natural components. Bottom panels show the individual Tier 1 source regions responsible for the long-range transported component of ozone.

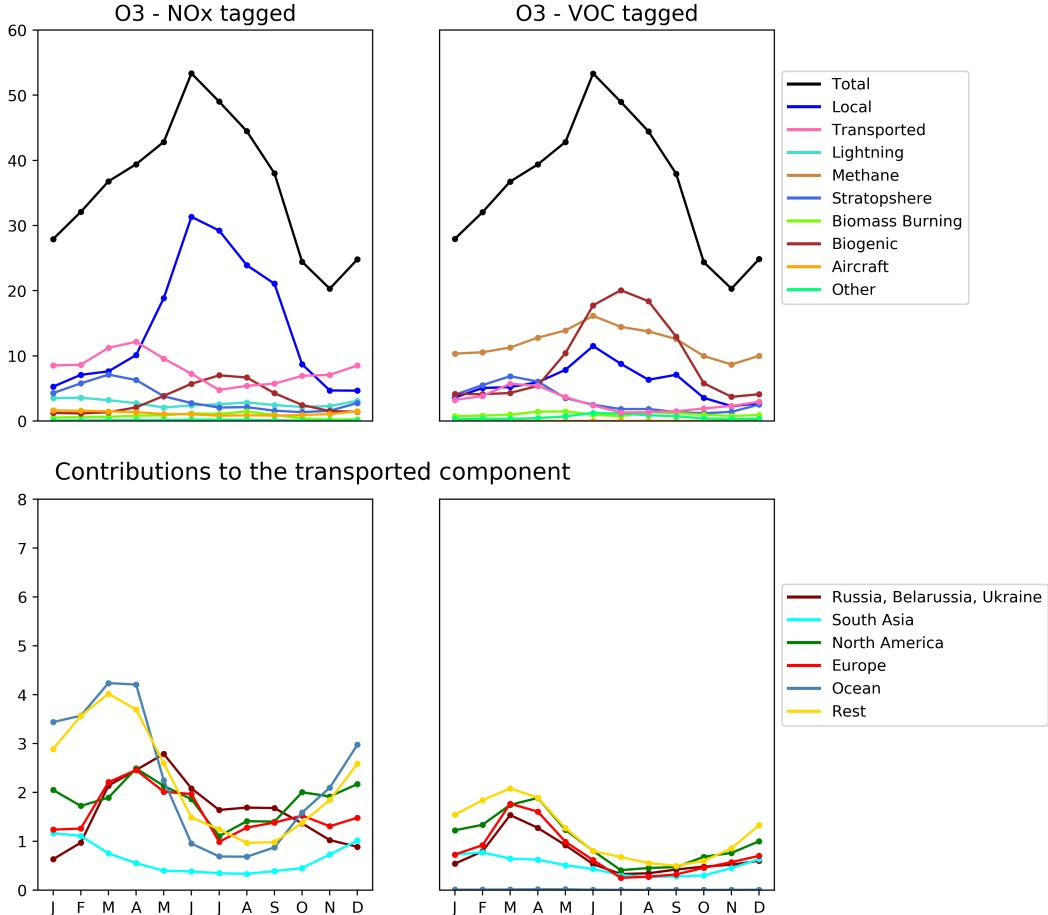

**Figure 6.** Seasonal cycle of surface ozone (ppb) in the HTAP Tier 2 receptor region "North East China". $NO_x$-tagging is shown in the left panels, and reactive carbon tagging in the right panels. Top panels show total monthly mean ozone (black line) as well as the local anthropogenic component, long-range transported anthropogenic component, and natural components. Bottom panels show the individual Tier 1 source regions responsible for the long-range transported component of ozone.

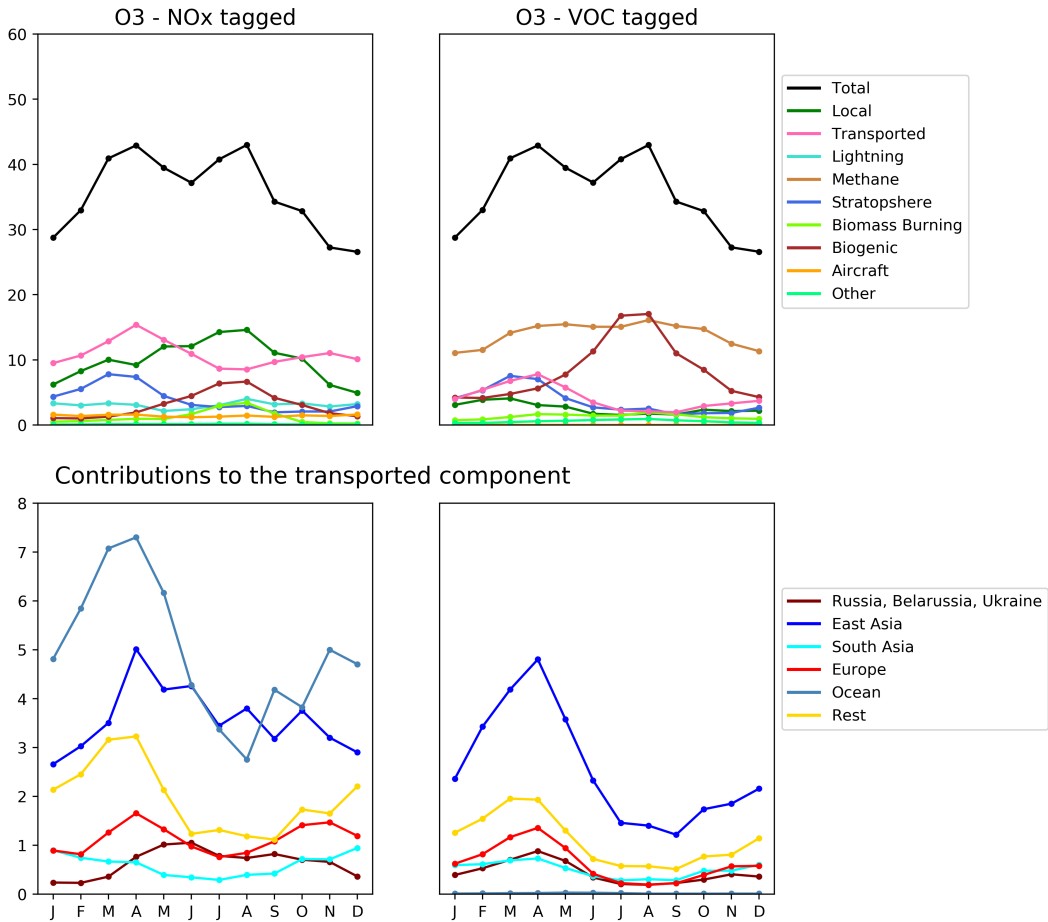

**Figure 7.** Seasonal cycle of surface ozone (ppb) in the HTAP Tier 2 receptor region "North West United States". $NO_x$-tagging is shown in the left panels, and reactive carbon tagging in the right panels. Top panels show total monthly mean ozone (black line) as well as the local anthropogenic component, long-range transported anthropogenic component, and natural components. Bottom panels show the individual Tier 1 source regions responsible for the long-range transported component of ozone.

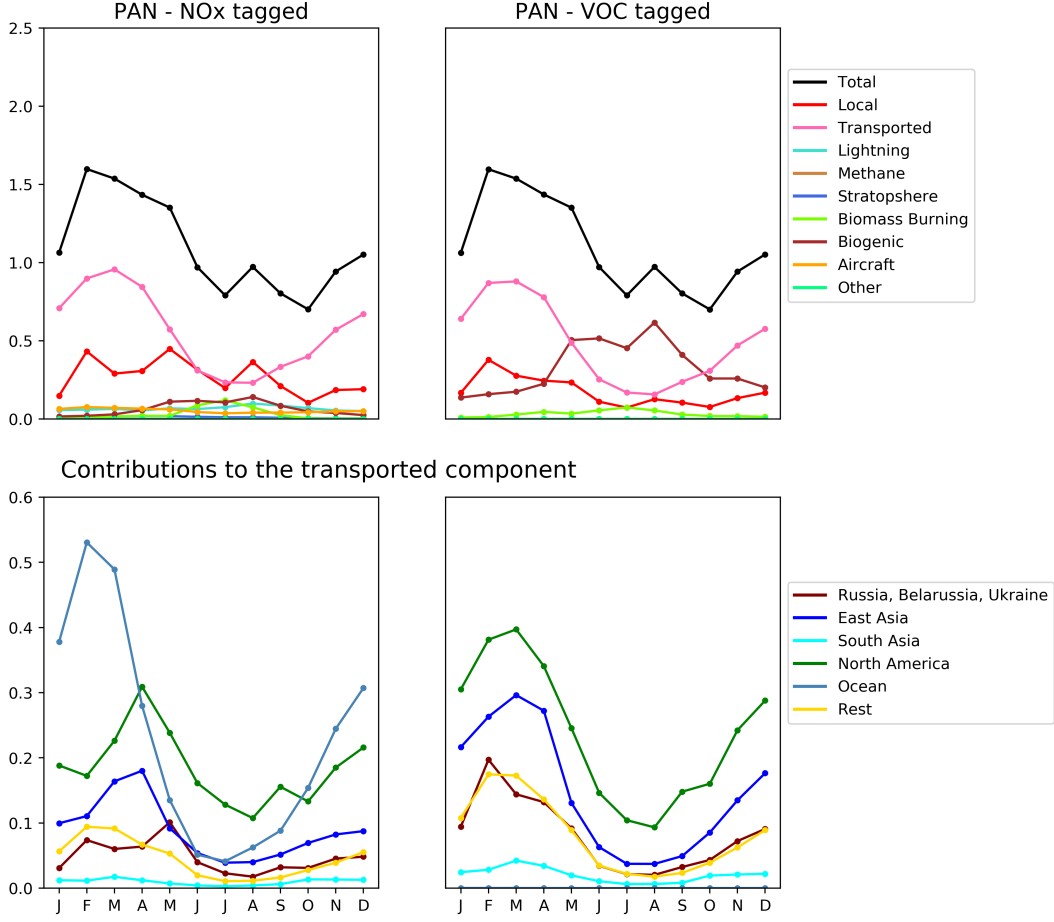

**Figure 8.** Seasonal cycle of column-integrated lower tropospheric PAN ($10^{-15}$ molec cm$^{-2}$) in the HTAP Tier 2 receptor region "North West Europe". The lower troposphere is defined here as all model levels between 800 and 500 hPa. NO$_x$-tagging is shown in the left panels, and reactive carbon tagging in the right panels. Top panels show total monthly mean PAN (black line) as well as the local anthropogenic component, long-range transported anthropogenic component, and natural components. Bottom panels show the individual Tier 1 source regions responsible for the long-range transported component of PAN.

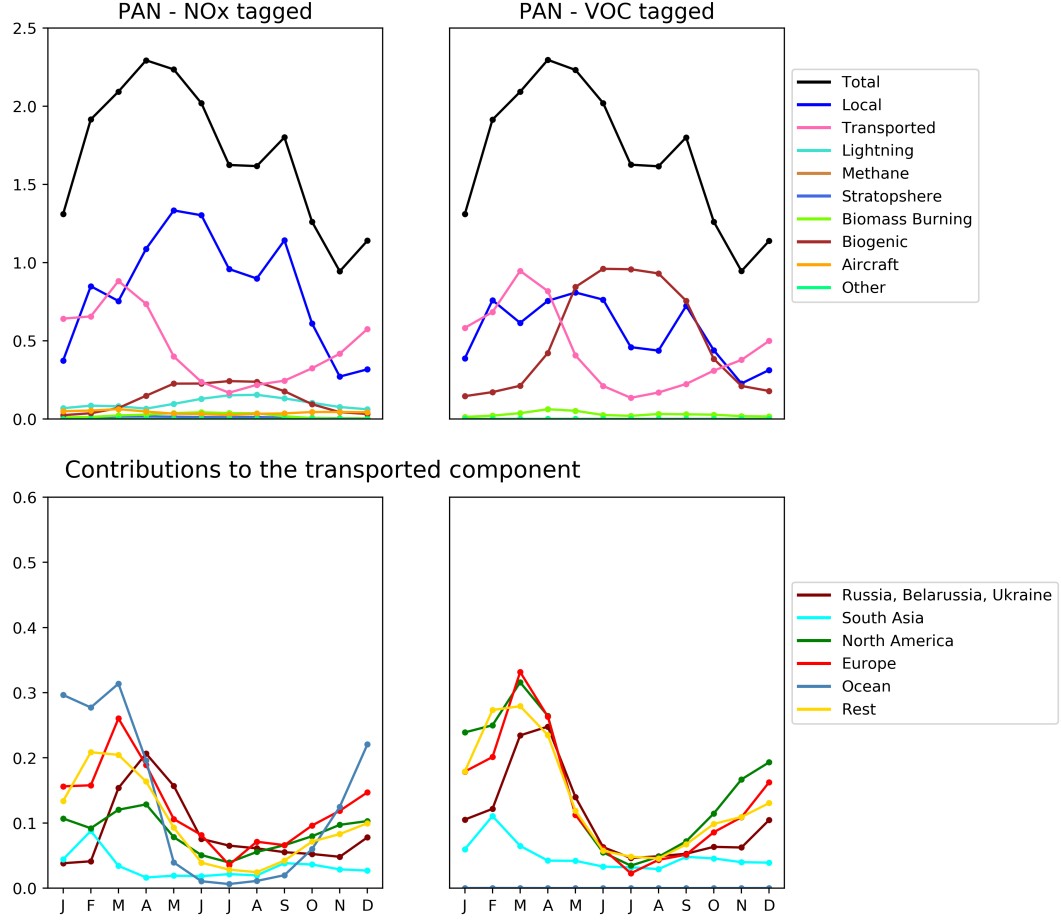

**Figure 9.** Seasonal cycle of column-integrated lower tropospheric PAN ($10^{-15}$ molec cm$^{-2}$) in the HTAP Tier 2 receptor region "North East China". The lower troposphere is defined here as all model levels between 800 and 500 hPa. NO$_x$-tagging is shown in the left panels, and reactive carbon tagging in the right panels. Top panels show total monthly mean PAN (black line) as well as the local anthropogenic component, long-range transported anthropogenic component, and natural components. Bottom panels show the individual Tier 1 source regions responsible for the long-range transported component of PAN.

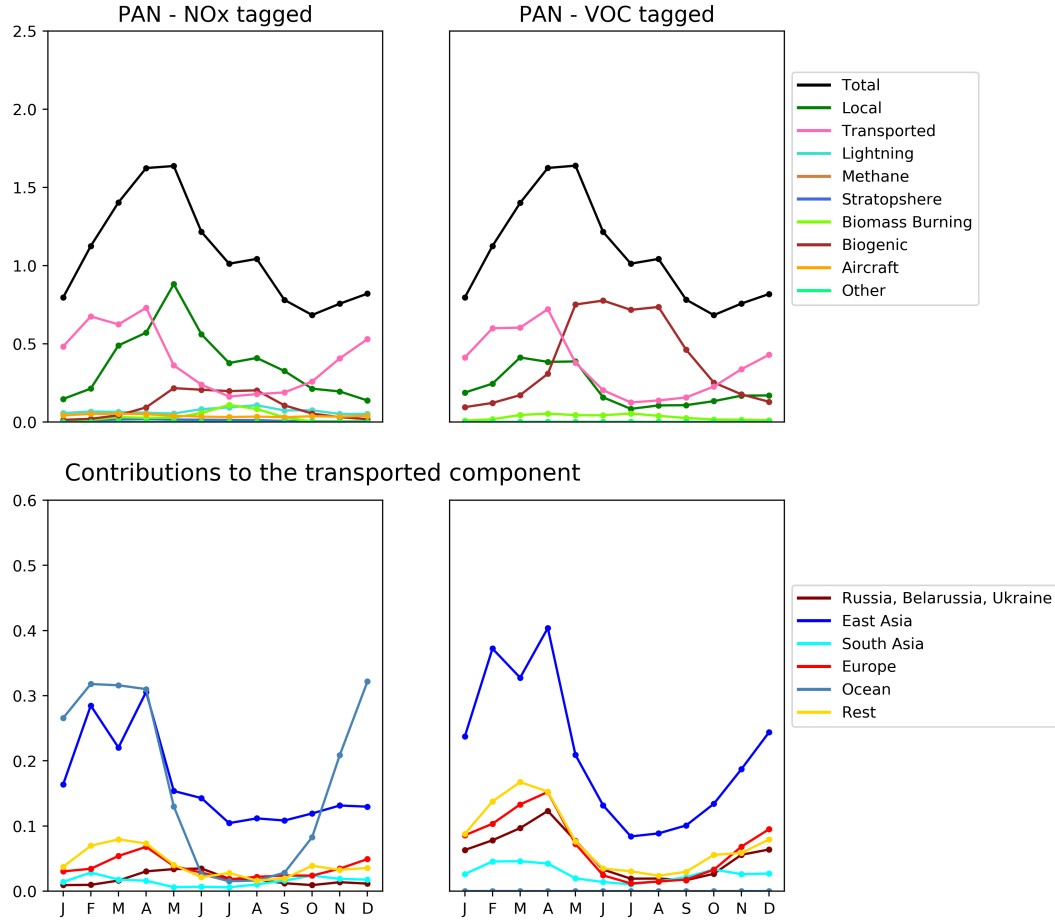

**Figure 10.** Seasonal cycle of column-integrated lower tropospheric PAN ($10^{-15}$ molec cm$^{-2}$) in the HTAP Tier 2 receptor region "North West United States". The lower troposphere is defined here as all model levels between 800 and 500 hPa. NO$_x$-tagging is shown in the left panels, and reactive carbon tagging in the right panels. Top panels show total monthly mean PAN (black line) as well as the local anthropogenic component, long-range transported anthropogenic component, and natural components. Bottom panels show the individual Tier 1 source regions responsible for the long-range transported component of PAN.

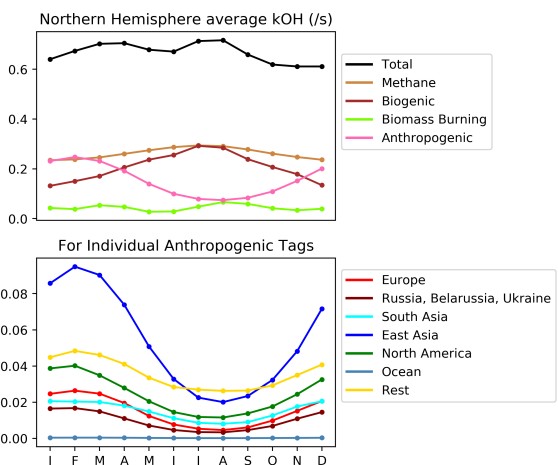

**Figure 11.** Seasonal cycle of northern hemispheric tropospheric column-integrated OH reactivity ($s^{-1}$) due to reactive carbon from the VOC-tagged run. The complete attribution is shown in the top panel, and the detailed attribution to anthropogenic emissions from HTAP Tier 1 source regions is shown in the bottom panel.

**Table 1.** List of tags used for attribution of tropospheric ozone in the $NO_x$- and VOC-tagged runs. Anthropogenic emissions of $NO_x$ and reactive carbon are tagged based on their HTAP Tier 1 region. Other tags are as in Butler et al. (2018).

| Tag name | $NO_x$-tagging | VOC-tagging |
|---|---|---|
| HTAP Tier 1 regions | | |
| Oceans[1] | Explicit | Explicit |
| North America | Explicit | Explicit |
| Europe | Explicit | Explicit |
| South Asia | Explicit | Explicit |
| East Asia | Explicit | Explicit |
| Russia, Belarus, Ukraine | Explicit | Explicit |
| South East Asia | Explicit | RoW |
| Northern Africa | Explicit | RoW |
| Middle East | Explicit | RoW |
| Middle America | Explicit | RoW |
| Central Asia | Explicit | RoW |
| Pacific, Australia, New Zealand | RoW | RoW |
| Southern Africa | RoW | RoW |
| South America | RoW | RoW |
| Arctic | RoW | RoW |
| Antarctic | RoW | RoW |
| Other tags | | |
| Stratosphere | Global[2] | Global |
| Aircraft | Global | Global |
| Biogenic | Global | Global |
| Biomass burning | Global | Global |
| Lightning | Global | N/A |
| Methane | N/A | Global |
| Extra production | Global[3] | Global[4] |

[1] $NO_x$ from "Oceans" is exclusively from shipping, while reactive carbon from this region is predominantly biogenic.

[2] For $NO_x$ tagging, the stratosphere tag is applied directly to ozone produced in the stratosphere (as for VOC tagging) and also to NO produced from dissociation of $N_2O$.

[3] For $NO_x$ tagging, "extra production" of ozone is due to the self reaction of OH radicals and reactions between $HO_2$ and organic peroxy radicals.

[4] For VOC tagging, "extra production" of ozone is due to the self reaction of OH radicals and reaction of OH with $H_2O_2$.

**Table 2.** Attribution of ozone to tagged sources of $NO_x$. Also shown is the contribution of the stratosphere, and the contribution of minor chemical production pathways in the troposphere. Contributions of each tagged source are shown to both the 2010 annual average tropospheric burden and to the annual average Northern Hemisphere surface mixing ratio. Where applicable, the Ozone Production Efficiency (OPE) of $NO_x$ emissions from each tagged source is also given.

| $NO_x$ Source | Emissions (Tg(N)/yr) | Ozone burden (Tg) | OPE (mol/mol) | NH surface (ppb) |
|---|---|---|---|---|
| Lightning | 3.43 | 80.5 | 6.85 | 3.14 |
| Stratosphere | – | 75.5 | – | 3.17 |
| Biogenic | 5.04 | 26.0 | 1.50 | 2.57 |
| Oceanic sources | 4.28 | 19.9 | 1.35 | 5.33 |
| East Asia | 9.97 | 16.9 | 0.495 | 3.01 |
| South East Asia | 1.62 | 15.3 | 2.76 | 0.755 |
| Aircraft | 0.646 | 12.2 | 5.49 | 1.13 |
| Biomass burning | 5.03 | 12.1 | 0.704 | 1.45 |
| South Asia | 3.49 | 10.8 | 0.907 | 1.27 |
| North America | 4.79 | 10.4 | 0.632 | 2.88 |
| Middle America | 1.27 | 8.81 | 2.02 | 1.00 |
| Europe | 3.16 | 4.81 | 0.444 | 1.76 |
| Middle East | 1.82 | 4.11 | 0.659 | 1.02 |
| RUS/BEL/UKR | 1.37 | 2.15 | 0.457 | 0.852 |
| North Africa | 0.531 | 1.72 | 0.947 | 0.513 |
| Central Asia | 0.287 | 0.627 | 0.638 | 0.238 |
| Rest of World | 2.54 | 15.7 | 1.80 | 0.500 |
| Extra production | – | 1.47 | – | 0.132 |
| Total trop. ozone | – | 319 | – | 30.7 |

**Table 3.** Attribution of ozone to tagged sources of reactive carbon. Also shown is the contribution of the stratosphere, and the contribution of minor chemical production pathways in the troposphere. Contributions of each tagged source are shown to both the 2010 annual average tropospheric burden and to the annual average Northern Hemisphere surface mixing ratio. Where applicable, the Ozone Production Efficiency (OPE) of reactive carbon emissions from each tagged source is also given.

| Reactive carbon source | Emissions VOC (Tg(C)/yr) | CO (Tg(C)/yr) | Ozone burden (Tg) | OPE (mol/mol(C)) | NH surface (ppb) |
|---|---|---|---|---|---|
| Methane | 410 | – | 113 | 0.0689 | 12.4 |
| Biogenic | 668 | 42.2 | 76.8 | 0.0270 | 7.20 |
| Stratosphere | – | – | 66.8 | – | 2.91 |
| Biomass burning | 29.8 | 162 | 13.4 | 0.0176 | 1.25 |
| East Asia | 20.0 | 80.4 | 10.3 | 0.0257 | 1.96 |
| South Asia | 16.2 | 36.5 | 6.67 | 0.0316 | 0.715 |
| North America | 12.0 | 23.8 | 4.39 | 0.0307 | 1.18 |
| Europe | 6.12 | 11.6 | 2.04 | 0.0288 | 0.695 |
| RUS/BEL/UKR | 4.04 | 4.92 | 1.22 | 0.0341 | 0.433 |
| Oceanic sources | 11.0 | 0.587 | 0.0957 | 0.00206 | 0.0162 |
| Rest of World | 55.3 | 82.8 | 19.5 | 0.0352 | 1.43 |
| Extra production | – | – | 4.58 | – | 0.546 |
| Total trop. ozone | – | – | 319 | – | 30.7 |

**Table 4.** Change in the contribution of reactive carbon sources to tropospheric and Northern Hemisphere surface ozone in response to a 25% increase in the imposed surface mixing ratio of methane. Absolute changes and percentage changes are both shown.

| Ozone source | Tropospheric Burden | | NH Surface mixing ratio | |
|---|---|---|---|---|
| | Change in ozone burden (Tg) | Percentage change | Change in mixing ratio (ppb) | Percentage change |
| Methane | 13.0 | 13.0 | 1.47 | 13.5 |
| Biogenic | -1.88 | -2.40 | -0.168 | -2.28 |
| Stratosphere | -0.683 | -1.01 | -0.0226 | -0.770 |
| Rest of World | -0.379 | -1.91 | -0.0315 | -2.15 |
| Biomass burning | -0.243 | -1.78 | -0.0316 | -2.47 |
| East Asia | -0.238 | -2.26 | -0.0583 | -2.90 |
| South Asia | -0.131 | -1.93 | -0.0155 | -2.13 |
| Extra production | -0.0351 | -0.761 | -0.00890 | -1.60 |
| North America | -0.0975 | -2.17 | -0.0351 | -2.88 |
| Europe | -0.0479 | -2.29 | -0.0222 | -3.09 |
| RUS/BEL/UKR | -0.0286 | -2.29 | -0.0132 | -2.96 |
| Oceanic sources | -0.00287 | -2.91 | -0 | -4.72 |
| Aircraft | -0.00158 | -2.59 | -0 | -3.81 |
| Total trop. ozone | 9.22 | 2.98 | 1.07 | 3.59 |

**Table 5.** Change in the contribution of $NO_x$ sources to tropospheric and Northern Hemisphere surface ozone in response to a 25% increase in the imposed surface mixing ratio of methane. Absolute changes and percentage changes are both shown.

| Ozone source | Tropospheric Burden | | NH Surface mixing ratio | |
| --- | --- | --- | --- | --- |
| | Change in ozone burden (Tg) | Percentage change | Change in mixing ratio (ppb) | Percentage change |
| Lightning | 3.37 | 4.37 | 0.149 | 4.97 |
| Stratosphere | -0.0598 | -0.0791 | 0.00980 | 0.310 |
| Biogenic | 0.915 | 3.65 | 0.0861 | 3.47 |
| Oceanic sources | 1.04 | 5.53 | 0.287 | 5.70 |
| East Asia | 0.532 | 3.25 | 0.0884 | 3.02 |
| Rest of World | 0.557 | 3.68 | 0.0167 | 3.45 |
| South East Asia | 0.509 | 3.43 | 0.0214 | 2.92 |
| Aircraft | 0.558 | 4.81 | 0.0557 | 5.20 |
| Biomass burning | 0.277 | 2.34 | 0.0361 | 2.55 |
| South Asia | 0.395 | 3.78 | 0.0370 | 3.00 |
| North America | 0.340 | 3.38 | 0.0928 | 3.33 |
| Middle America | 0.313 | 3.68 | 0.0309 | 3.19 |
| Europe | 0.154 | 3.31 | 0.0559 | 3.29 |
| Middle East | 0.146 | 3.67 | 0.0389 | 3.96 |
| RUS/BEL/UKR | 0.0703 | 3.38 | 0.0269 | 3.26 |
| North Africa | 0.0736 | 4.46 | 0.0231 | 4.72 |
| Extra production | 0.00229 | 0.156 | -0 | -0.00590 |
| Central Asia | 0.0235 | 3.89 | 0.00936 | 4.09 |
| Total trop. ozone | 9.22 | 2.98 | 1.07 | 3.59 |