# Peer review of "Attribution of ground-level ozone to anthropogenic and natural sources of $NO_x$ and reactive carbon in a global chemical transport model"

_Atmospheric Chemistry and Physics, 2020_

## Author Comment (AC1) · 19 Jun 2020

Please note that the manuscript has been updated on June 19. A mistake in the ordering of Figures 5-10 has been corrected. The mistake occurred during the final stages of preparing the manuscript for submission. The figures themselves, the figure captions, and any related discussion in the body of the manuscript were not affected in any way by this mistake. The graphics files were simply included in the wrong order.

The version currently online contains all graphics files included in the correct order. On behalf of all co-authors I would like to apologise for any confusion this may have caused.

[Figure]

Tim Butler

---

## Referee Comment (RC1) · Anonymous Referee #1 · 24 Jun 2020

Summary: Authors use TOAST in CAM-Chem to attribute ozone to NOx, VOC, or methane separately. Thus, each molecule of ozone is attributed either 100% to NOx, 100% to VOC or 100% to methane. This method likely overstates the role of non-NOx molecules at the global scale because of primarily NOx-limited environments, but because both are showed, this is not a limitation. The results highlight the importance of long-range transport to annual averages, the disproportionate affect of the global shipping sector, and the shipping-methane interactions.

Response: The paper is a nice contribution that is well written with mostly minor recommendations from this reviewer. The comments will primarily be shown in the line by

line, but what follows characteristic of my response.

A few scattered recommendations are summarized here. The citations often seem inappropriately recent for well-established phenomenon. There are few qualitative statements without numerical context. The authors focus on selected regions, but provide no specific reason these were chosen nor do they contrast the results with the general case.

The ozone production efficiency metrics and discussion may require more edits. The authors compare OPE between methane and VOC uses table numbers that are in mol/mol, but in the text they say mol/molC. This is particularly problematic because they then highlight the high efficiency of methane. The OPE definition also seems inconsistent in the text compared to how it appears to be calculated. Further detail on the OPE response is in the line-by-line comments.

The authors highlight shipping, yet figures and discussion rarely figures and tables say "Ocean" which according to Table 1includes both natural and anthropogenic sources. This has two major implications for the paper. First, I am assuming that "Ocean NOx" is being implicitly assumed to be all from shipping. This makes sense as an assumption, but should be explicitly stated. For example, the soil NOx emissions from CAMS have an over ocean component which would violate this assumption. Second, ocean is treated as long-range transport and/or extra-regional. Figure 2 suggests that shipping has a low inland penetration that might suggest strongest influences from nearby production. If shipping from state or federal waters is most influential, it may be inappropriate to label it extra-regional or long-range transport. This requires some clarification and perhaps applying more nuanced assignment of "ocean/shipping." Finally, making the connection clear and explicit in your discussion would help readers.

These comments can largely be addressed by textual changes that should mostly be easy to implement.

Line-by-line:

17 - Monks et al. 2015 is a very recent citation for such a well-known phenomenon.

18 - Fleming et al., 2018 is a very recent citation for harm to health. Mills et al. 2018 is a very recent citation for harm to vegetation. There is a long history include protective legislation for decades for both of these.

26 - I think baseline has not yet been defined.

30 to 31 - Consider adding a reference to Cooper et al sonde evaluations or TOAR.

35 - "extremely high" should include a numerical context. What percent of the mean?

59 - Why is NOx from ships having more influence important?

156 - Consider citing a reason for 1760 ppb

159 - How was 2-years estabilished as sufficient? What was the spinup for the methane case?

216 - It is not clear to this reviewer that OPE was calculated based on gross production as described on this line. See comments on lines 284 to 285.

218 - "some source regions" should be enumerated.

271 to 273 - This is a long established phenomenon, which if often expressed as yield of RO2 per mole. This includes books from the 1990s. A 2011 citation seems recent.

284 to 285 - Table 3 reports OPE as mol/mol but mol/molC is in the text. Methane, ocean, and biogenic are dominated by VOC over CO. It appears that OPE is being calculated by converting emissions to molC/yr and burden ozone to mol. For methane, the OPE calcualted this way is identical. For ocean and biogenic, which have some CO, the OPE is nearly identical to your value. This makes me think that the OPE is in mol/molC, not mol/mol. This has consequences for the way results are discussed. For example, assuming that most VOC mass has 4 carbons, the OPEs for NMVOCs increase by 4x. So comparing CH4 to on a mol/molC basis seems odd.

On the same topic, this method of OPE calculation is different than what this reviewer is used familiar with or as you described on line 216. Your line 216 is more consistent with Kleinman 2002, who cites Liu 1987 and Lin 1988 to define OPE as "the number of molecules of oxidant ($O_3$ + $NO_2$) produced photochemically when a molecule of $NO_x$ (NO + $NO_2$) is oxidized." Thus, OPE would be related to gross production not burden. Burden is a net state, which includes both production and loss. Because anthropogenic VOCs react near the surface, they may be subject to higher deposition loss rates and shorter chemical lifetimes. Thus, an OPE based on state rather than production, may weigh in methane's favor.

The indicator you are using, regardless of the name, is clearly useful. The definition and discussion needs to be adjusted to better match what you have done, and to make a better comparison between moles of NMVOC and moles methane.

299 - source attribution includes zero-out/perturbation techniques, while tagging does not. It might be worth using specific language here. While I am not aware of reactive carbon zero-out/perturbation techniques, there is a lot of literature out there.

322 - "natural sources and long-range transport ... [each or together] contribute more to" It was unclear if this was a combined statement.

325 to 326 - This is a complex statement. First, excluding "Ocean" this is not true for NAM. Then, the question is does shipping have a far reaching effect or is it localized. If it is localized and within the regions Exclusive Economic Zone (potentially even within state waters), then is it "intra-regional" or "extra-regional."

337 - A select few regions are shown, but no explanation of why they are shown is provided. Are the typical sites or the sites where transport matters most?

355 to 356 - For China and USA, the minimum contribution is not in winter.

365 - In all regions or in all three regions shown?

366 - The "pronouncement" of the cycle is not clearly stronger in all regions even

though it commonly is.

418 - Did you study fate? loss processes?

428 - Generally associated with PAN discussion (not really line specific). How do you treat the equilibrium reactions that are often artificially defined as "net" forward rates? What implications does that have for tagging approaches to look at PAN?

434 to 464 - Is this mean to help explain PAN or be a separate discussion? The OH reactivity doesn't consider the PAN potential, which is related to the ability to create a peroxy acyl radical. If this is a new thought, perhaps add some sort of transition.

458 - What level of confidence do you have in the Asian VOC? How are VOC speciated differently by region? This could have more general implications in other places in the paper.

480 - Given the NOx-limitation changes due to removing methane, how does assuming linearity in inverting the perturbation?

482 - spinup length for methane?

492 - gross production or net burden?

492 to 496 - I found the relative increases as ambiguous. Increases in total or increases? in methane direct? or increase in net methane contribution? Ultimately, I found the table more clear than the discussion.

499 to 500 - might note this is for annual averages

568 - and stratosphere.

---

## Referee Comment (RC2) · Volker Grewe (Referee) · 9 Jul 2020

Butler et al. present an analysis of modelled surface ozone concentrations with respect to the chemical production via either NOx or VOCs. The paper is well written and offers important insights in the relation between regional emissions and ozone surface mixing ratios. However, I think some more comments on

- Interpretation of the diagnostics and

- Uncertainties

should be given.

Interpretation:

a) Loss processes

As far as I understood the ozone production terms are taken into account in the tagging scheme for ozone. How is the ozone destruction treated? Increase in the NOx emissions and hence NOx concentrations affect not only ozone production, but also the lifetime of ozone (e.g. Stevenson et al. 2006). Hence also the individual sources contribute differently to the ozone destruction. How would your results change, if you take this effect into account?

b) Ambiguity

While the separation of the ozone production wrt NOx and VOC is very helpful in understanding the driving mechanisms, it may also appear as ambiguous. E.g. Figure 4 indicates that European ozone is largely dominated by NOx from ozone (top) or methane (bottom). That sounds like a contradiction. Shouldn't it be in the end one ozone bar having all contributions included, instead of two (top and bottom figure)? I think it would be helpful to add some discussions here.

Uncertainty:

a) Resolution:

The plume processes for ships are mentioned, which I think is an important process to be considered. But what is about model resolution in general? Does this affect city or harbour plumes as well?

b) Quality of emission data

How sensitive are the results to uncertainties from emission data. Biogenic emissions, etc. ?

Minor Comments:

page 2 / line 53/54 Dahlmann et al. calculated explicitly the ozone production efficiency and showed that lightning and aviation NOx emissions are most efficient, in case you want to quantify the number of ozone molecules per emitted NOx.

page 4 / line 108 Grewe (2013) provided a theoretical framework for taking into account these competing effects and compared that in a simple framework in Grewe et al (2010) and in a chemistry-climate model in Grewe et al. (2017). Please rephrase that this is NOT common to all tagging schemes.

line 115: There is also a nice table in recently published work by Mertens et al. (2020) (https://doi.org/10.5194/acp-20-7843-2020) discussing in detail the characteristics of these two methods, which might be helpful here.

line 115: Perhaps you want to adapt the naming consistently throughout the manuscript? contribution/share for tagging approaches changes/impact for perturbation? E.g. line 56 Hoor et al used perturbation approaches. The wording "contribution" in this respect might be misguiding.

line 126: This statement might be misunderstood. The Butler et al tagging scheme is the only one, which separately analyses attribution of tropospheric ozone to its NOx and reactive carbon precursors, whereas the Grewe et al scheme is the only, which analysis attribution of tropospheric ozone to both together, NOx and reactive carbon precursors, taking competing effects into account.

line 159: Is there any reason why the spin-up differs?

line 233: also Dahlmann et al. 2011.

line 401: Please elaborate a little bit more on this comparison in terms of quantitative values. I guess we should expect a difference in the strength of the contribution vs perturbation? If so, can that be explained by the difference in the method?

line 550: RF and human health effects are not calculated. Please re-phrase that this is a potential important impact based on literature and not your findings.

[Figure]

Figure 4: Please adapt the text in the figure to explicitly state that surface ozone is presented. Caption: Please include some more details, e.g. "Source-receptor relationships between annual averaged surface ozone volume mixing ratio and NOx and VOC emission type and region.", in order to clarify that with region the NOx emission and not the ozone production is meant.

---

## Author Comment (AC2) · 21 Jul 2020

**Response to referee comments on acp-2020-436**

Tim Butler[1,2], Aurelia Lupascu[1], and Aditya Nalam[1,2]

[1]Institute for Advanced Sustainability Studies, Potsdam, Germany
[2]Institut für Meteorologie, Freie Universität Berlin, Germany

We thank Volker Grewe and the Anonymous Referee for their comments on our manuscript. Our responses are given below. In each case we reproduce the referee comments in boldface, and provide our responses in standard script. Where we have made changes to the original manuscript in response to the referee comments, these changes are described together with our response. Both referees noted that the manuscript is well written. In our modifications to the manuscript we have tried to balance preservation of as much of the original text as possible, with what we hope are changes that satisfy the comments of the referees. We believe that the revised version of the manuscript has improved compared to the original, and thank the referees again for their comments. In our responses we refer to line numbers in the original manuscript. We append a revised manuscript here with the changes mentioned in our responses marked up.

**Response to Anonymous Referee #1**

**Summary: Authors use TOAST in CAM-Chem to attribute ozone to NOx, VOC, or methane separately. Thus, each molecule of ozone is attributed either 100% to NOx, 100% to VOC or 100% to methane. This method likely overstates the role of non- NOx molecules at the global scale because of primarily NOx-limited environments, but because both are showed, this is not a limitation. The results highlight the importance of long-range transport to annual averages, the disproportionate affect of the global shipping sector, and the shipping-methane interactions.**

We would like to correct an apparent misunderstanding by the referee here. Our source attribution method does not perform a 100% attribution of ozone to methane. Rather, methane is treated as just one form of the many forms of "reactive carbon", to which we do perform a 100% attribution (in addition to the 100% attribution to $NO_x$). While we do give a definition of reactive carbon in the paragraph beginning on line 67, we have also added text to the first paragraph of the introduction section to make it even clearer that reactive carbon includes methane.

**Response: The paper is a nice contribution that is well written with mostly minor recommendations from this reviewer. The comments will primarily be shown in the line by line, but what follows characteristic of my response.**

**A few scattered recommendations are summarized here. The citations often seem inappropriately recent for well-established phenomenon. There are few qualitative statements without numerical context. The authors**

focus on selected regions, but provide no specific reason these were chosen nor do they contrast the results with the general case.

The ozone production efficiency metrics and discussion may require more edits. The authors compare OPE between methane and VOC uses table numbers that are in mol/mol, but in the text they say mol/molC. This is particularly problematic because they then highlight the high efficiency of methane. The OPE definition also seems inconsistent in the text compared to how it appears to be calculated. Further detail on the OPE response is in the line-by-line comments.

The authors highlight shipping, yet figures and discussion rarely figures and tables say "Ocean" which according to Table 1includes both natural and anthropogenic sources. This has two major implications for the paper. First, I am assuming that "Ocean NOx" is being implicitly assumed to be all from shipping. This makes sense as an assumption, but should be explicitly stated. For example, the soil NOx emissions from CAMS have an over ocean component which would violate this assumption. Second, ocean is treated as long-range transport and/or extra-regional. Figure 2 suggests that shipping has a low inland penetration that might suggest strongest influences from nearby production. If shipping from state or federal waters is most influential, it may be inappropriate to label it extra-regional or long-range transport. This requires some clarification and perhaps applying more nuanced assignment of "ocean/shipping." Finally, making the connection clear and explicit in your discussion would help readers.

These comments can largely be addressed by textual changes that should mostly be easy to implement.

We have taken note of these general comments, and provide our responses as appropriate after the line-by-line comments from the referee where the relevant specific comments are made. In response to the point about the "Ocean" region, we have added a sentence to the paragaph discussing Table 1 to make it clear that all $NO_x$ from this region is due to shipping, and that the predominant source of reactive carbon is biogenic DMS. We have also modified the relevant footnote in Table 1 to repeat this point.

**17 - Monks et al. 2015 is a very recent citation for such a well-known phenomenon.**

The referee is correct. We have generally tried to provide recent references in order to direct interested readers to relatively new work, which itself usually contains further references, but we acknowledge that also including older references here can provide a fuller context. In this case, we have added the citation of Crutzen (1973) to the statement in question, and in the same spirit, added a citation of Atkinson (2000) later in the same paragraph.

**18 - Fleming et al., 2018 is a very recent citation for harm to health. Mills et al. 2018 is a very recent citation for harm to vegetation. There is a long history include protective legislation for decades for both of these.**

Again, we agree. We have added references to Haagen-Smit (1952) and Reich and Amundson (1985), which are both examples of important early work that are still well-cited today.

**26 - I think baseline has not yet been defined.**

Here we are actually already defining baseline to mean the long-range transported component of ozone. We have added a sentence to make this clearer, and given references to Parrish et al. (2017) and Derwent et al. (2018) which both use the term this way.

**30 to 31 - Consider adding a reference to Cooper et al sonde evaluations or TOAR.**

We have discussed TOAR (and used data from the project) later in the manuscript. In the text in question here, the point we are making is about recent trends in two specific regions, which we think is already adequately made with the references we have included, so we prefer to leave the manuscript unchanged here.

**35 - "extremely high" should include a numerical context. What percent of the mean?**

We show this spread in our Figure 1, and comment on its magnitude. To respond to the this comment, we have added some text to the discussion of Figure 1 at the beginning of Section 3 noting that the spread in the model estimates is of a similar order to the Northern Hemisphere annual mean surface ozone itself, and also added a forward to reference to this section to the part of the introduction mentioned by the referee.

**59 - Why is NOx from ships having more influence important?**

This statement about earlier results is included to indicate that our study is not the first to point out that ship $NO_x$ has more influence on surface ozone than aircraft $NO_x$, despite the latter being more efficient at producing ozone. We feel that this is worth keeping in the manuscript.

**156 - Consider citing a reason for 1760 ppb**

This value comes from the model setup that we adapted from Tilmes et al. (2015) for this work. We have clarified this in the manuscript text.

**159 - How was 2-years estabilshed as sufficient? What was the spinup for the methane case?**

As mentioned in our earlier response to this referee, there is no "methane case" in this manuscript. We hope that our earlier response has clarified that. The model spinup for the $NO_x$ and reactive cases is described in more detail in Butler et al. (2018), which is referenced twice in the paragraph in question. In order to respond to this comment, we added some extra text in response to this comment, noting that the model was deemed to be spun up when the maximum difference in December mean surface ozone attributable to any tagged source was less than 1% in any two subsequent years of simulation.

**216 - It is not clear to this reviewer that OPE was calculated based on gross production as described on this line. See comments on lines 284 to 285.**

We agree with the referee that our definition of OPE is unclear. We have clarified our definition of this quantity in the manuscript text to indicate that we define this as the contribution of each emission source to the tropospheric ozone burden, with units of moles of ozone per mole of N or C emitted.

**218 - "some source regions" should be enumerated.**

We have added that specifically South East Asia, Northern Africa, the Middle East, Middle America, and Central Asia are included with the "Rest of the World" in Figure 2 in order to enable better comparison Figure 3.

**271 to 273 - This is a long established phenomenon, which if often expressed as yield of RO2 per mole. This includes books from the 1990s. A 2011 citation seems recent.**

We have added references to Bowman and Seinfeld (1994) and Atkinson (2000).

**284 to 285 - Table 3 reports OPE as mol/mol but mol/molC is in the text. Methane, ocean, and biogenic are dominated by VOC over CO. It appears that OPE is being calculated by converting emissions to molC/yr and burden ozone to mol. For methane, the OPE calcualted this way is identical. For ocean and biogenic, which have some CO, the OPE is nearly identical to your value. This makes me think that the OPE is in mol/molC, not mol/mol. This has consequences for the way results are discussed. For example, assuming that most VOC mass has 4 carbons, the OPEs for NMVOCs increase by 4x. So comparing CH4 to on a mol/molC basis seems odd.**

**On the same topic, this method of OPE calculation is different than what this reviewer is used familiar with or as you described on line 216. Your line 216 is more consistent with Kleinman 2002, who cites Liu 1987 and Lin 1988 to define OPE as "the number of molecules of oxidant (O3 + NO2) produced photochemically when a molecule of NOx (NO + NO2) is oxidized." Thus, OPE would be related to gross production not burden. Burden is a net state, which includes both production and loss. Because anthropogenic VOCs react near the surface, they may be subject to higher deposition loss rates and shorter chemical lifetimes. Thus, an OPE based on state rather than production, may weigh in methane's favor.**

**The indicator you are using, regardless of the name, is clearly useful. The definition and discussion needs to be adjusted to better match what you have done, and to make a better comparison between moles of NMVOC and moles methane.**

We have updated Table 3 to clarify that the OPE of reactive carbon is reported in mol/mol(C), and we have made some small changes to the paragraph beginning on line 284 to consistently report the OPE in its correct units, and to reinforce that the OPE is "per unit of reactive carbon". Additional related changes were also made elsewhere in the manuscript based on an earlier comment by this referee.

We believe that the normalisation of the contribution of each VOC molecule to the tropospheric ozone burden by its carbon content does make sense in a global model, in which emitted species have the time to be completely broken down by photochemical and other loss processes. More carbon per molecule means more chemical bonds, and potentially more opportunity

for a large molecule to generate the peroxy radicals which can go on to form ozone, even if these processes all take a lot more
time than the ozone production over regional scales, where the molar emissions and the OH rate constant are more important
at determining the ozone proudction. By removing this effect, we are able to discuss the OPE of reactive carbon in terms of the
amount of detail in the model chemical mecanism and the co-emission of $NO_x$. After careful consideration we do not see any
need to modify our discussion of OPE any further.

**299 - source attribution includes zero-out/perturbation techniques, while tagging does not. It might be worth
using specific language here. While I am not aware of reactive carbon zero-out/perturbation techniques,
there is a lot of literature out there.**

We stand by our statement, and note that neither referee has provided any references to earlier work which actually does
perform a complete attribution of tropospheric ozone to reactive carbon precursors alone. Furthermore, we note that the recent
review by Heald and Kroll (2020) cites only the work of Butler et al. (2018) as the source of data for its Figure 1, in which the
contribution of reactive carbon to tropospheric ozone is shown.

**322 - "natural sources and long-range transport ... [each or together] contribute more to" It was unclear if
this was a combined statement.**

Yes, this is intended to be a combined statement. We have added the word "together" to make this clearer.

**325 to 326 - This is a complex statement. First, excluding "Ocean" this is not true for NAM. Then, the ques-
tion is does shipping have a far reaching effect or is it localized. If it is localized and within the regions Exclu-
sive Economic Zone (potentially even within state waters), then is it "intra-regional" or "extra-regional."**

The question of the geographical extent of the effect of shipping emissions on surface ozone remains open. We have added
some extra text to the paragraph beginning on line 389 (original manuscript) indicating that future work should also include
more refined attribution of ozone to shipping emissions from coastal regions and the high seas.

**337 - A select few regions are shown, but no explanation of why they are shown is provided. Are the typical
sites or the sites where transport matters most?**

We have added a sentence immediately following this which gives some reasons for the selection of these regions.

**355 to 356 - For China and USA, the minimum contribution is not in winter.**

We are not sure what we originally intended to communicate with this sentence. We have simply deleted it, and do not believe
that this detracts from the discussion of our results in any meaningful way.

**365 - In all regions or in all three regions shown?**

This feature is seen in all regions, we have made this more explicit.

**366 - The "pronouncement" of the cycle is not clearly stronger in all regions even though it commonly is.**

We have deleted "but more pronounced".

150      **418 - Did you study fate? loss processes?**

We address this issue in our response to the review by Volker Grewe.

**428 - Generally associated with PAN discussion (not really line specific). How do you treat the equilibrium reactions that are often artificially defined as "net" forward rates? What implications does that have for tagging approaches to look at PAN?**

155      In our system, the forwards and backwards reactions are modelled explicitly. Butler et al. (2018) provides a complete description of the chemical mechanism and how it is tagged. We consider the implementation of our tagging scheme in additional chemical mechanisms as an open research area.

**434 to 464 - Is this mean to help explain PAN or be a separate discussion? The OH reactivity doesn't consider the PAN potential, which is related to the ability to create a peroxy acyl radical. If this is a new thought, perhaps add some sort of transition.**

160

The text in question is indeed distinct from the disussion of PAN, so we have added a new sub-section heading to aid the transition.

**458 - What level of confidence do you have in the Asian VOC? How are VOC speciated differently by region? This could have more general implications in other places in the paper.**

165      VOC speciation remains a challenge for emission inventories. The emission inventory used here is described by (Janssens-Maenhout et al., 2015) and specifies NMVOC as a total rather than with a speciation.

**480 - Given the NOx-limitation changes due to removing methane, how does assuming linearity in inverting the perturbation?**

The responses in receptor regions to reduction perturbations are often shown inverted (HTAP, 2010; Jonson et al., 2018, eg.).
170      We believe that this makes the results easier to follow.

**482 - spinup length for methane?**

As noted in our response to an earlier comment from the referee, there is no separate attribution of ozone to methane. We hope that our earlier response has clarified this.

**492 - gross production or net burden?**

175     The previous sentence already states clearly that we are referring to the net burden of ozone.

**492 to 496 - I found the relative increases as ambiguous. Increases in total or increases? in methane direct? or increase in net methane contribution? Ultimately, I found the table more clear than the discussion.**

We agree that the text here was unclear. We have tried to make it clearer when we are referring to the ozone attributable to methane using the tagging system, and when we are referring to the change in the ozone burden as a response to the change in
180     methane. We also switched to using the absolute change in ozone burden to introduce the point that the ozone increase due to rising methane is partialy offset by less ozone production from other VOC. We hope that these modifications have helped to improve the readability of this paragraph.

**499 to 500 - might note this is for annual averages**

We have clarified this statement so that it refers to annual average ozone.

185     **568 - and stratosphere.**

We have added a mention of the stratospheric contribution to springtime ozone to the beginning of this paragraph.

**Response to Volker Grewe**

**Butler et al. present an analysis of modelled surface ozone concentrations with respect to the chemical production via either NOx or VOCs. The paper is well written and offers important insights in the relation**
190     **between regional emissions and ozone surface mixing ratios. However, I think some more comments on**

      **– Interpretation of the diagnostics and**

      **– Uncertainties**

**should be given.**

**Interpretation:**

195     **a) Loss processes As far as I understood the ozone production terms are taken into account in the tagging scheme for ozone. How is the ozone destruction treated? Increase in the NOx emissions and hence NOx concentrations affect not only ozone production, but also the lifetime of ozone (e.g. Stevenson et al. 2006). Hence also the individual sources contribute differently to the ozone destruction. How would your results change, if you take this effect into account?**

200     It is not clear exactly which loss processes the referee is referring to here. One well-known effect of $NO_x$ emissions in polluted regions is the temporary removal of ozone often referred to as "titration". This loss of ozone is temporary, since the $NO_2$ it

produces can rapidly photolyse, ultimately yielding ozone again. Our tagging scheme accounts for this by tracking not just ozone, but all members of the "odd oxygen" family, as described in Butler et al. (2018).

More generally, tagged ozone (and other odd oxygen species) are lost at the same rate as the corresponding non-tagged species in the model. The tagging scheme thus delivers information on the contribution of different precursors to the modelled odd oxygen species (including ozone) at any given model time step. In this sense the method is similar to the method described by Grewe et al. (2017). Further details are given in Butler et al. (2018), which is referenced extensively from the current manuscript.

In cases where changing emissions might cause secondary effects in the lifetime of ozone or other odd oxygen species, these effects could be investigated using perturbation runs to explore the sensitivity of ozone to changes in the chemical environment. We have added some text to the introductory paragraph in which the tagging and perturbation approaches are compared to acknowledge this effect.

**b) Ambiguity While the separation of the ozone production wrt NOx and VOC is very helpful in understanding the driving mechanisms, it may also appear as ambiguous. E.g. Figure 4 indicates that European ozone is largely dominated by NOx from ozone (top) or methane (bottom). That sounds like a contradiction. Shouldn't it be in the end one ozone bar having all contributions included, instead of two (top and bottom figure)? I think it would be helpful to add some discussions here.**

Actually we believe that by separating the attribution of ozone into its two chemically distinct precursors, our method actually removes a lot of the ambiguity which could be caused if this were not done. For example, there are both anthropogenic and biogenic sources of both $NO_x$ and reactive carbon, but much more ozone is produced by the interaction of anthropogenic $NO_x$ with biogenic reactive carbon than the other way around. Without the separation of $NO_x$ and reactive carbon in the attribution scheme, this detail risks being lost. As an example of this, the recent study of Mertens et al. (2020) (mentioned by this referee below) did not distinguish between biogenic $NO_x$ and reactive carbon emissions, thus introducing ambiguity about the exact nature of the biogenic influence on ozone in their results.

Fundamentally, each ozone molecule produced chemically in the troposphere has two precursors, one $NO_x$, and one reactive carbon. Any source attribution system which does not distinguish between these two distinct types of precursors in some way must necessarily lose information about the origin of ozone. We don't see how we could possibly make this point any more clearly. It features prominently in the abstract, the first paragraph of the introduction, and in the structure of the manuscript itself.

**Uncertainty:**

**a) Resolution: The plume processes for ships are mentioned, which I think is an important process to be considered. But what is about model resolution in general? Does this affect city or harbour plumes as well?**

Yes, this process certainly does apply more generally than just to ship plumes. We have added a reference to Wild and Prather (2006) in the discussion of plume chemistry in order to bring this point out more.

**b) Quality of emission data How sensitive are the results to uncertainties from emission data. Biogenic emissions, etc. ?**

Uncertainties in emissions are a major issue in atmospheric chemistry modelling. Most emission inventories do not include uncertainty estimates, the inventory used in this study (Janssens-Maenhout et al., 2015) being no exception. The focus of our study is not the exploration of the sensitivity to uncertainties in emissions, but rather the application of our still relatively new tagging method to understand ozone source/receptor relationships using a well-established emission inventory. Specifically with regard to emissions, our study highlights the role of shipping, and of NMVOC emissions from East Asia as potentially important for further investigation in future work. This is one of the strengths of the tagging approach in general, which has already been pointed out by Grewe et al. (2010).

**Minor Comments:**

**page 2 / line 53/54 Dahlmann et al. calculated explicitly the ozone production efficiency and showed that lightning and aviation NOx emissions are most efficient, in case you want to quantify the number of ozone molecules per emitted NOx.**

We have added a reference to Dahlmann et al. (2011).

**page 4 / line 108 Grewe (2013) provided a theoretical framework for taking into account these competing effects and compared that in a simple framework in Grewe et al (2010) and in a chemistry-climate model in Grewe et al. (2017). Please rephrase that this is NOT common to all tagging schemes.**

We have replaced the word "problem" here with "challenge", in order to avoid creating the impression that other tagging schemes are somehow deficient or incorrect. But we stand by the fundamental point that tropospheric ozone is primarily produced through interaction of two chemically distinct types of precursors, and that this does indeed create a challenge for ozone source attribution. Different methods make different choices about how they meet this challenge, which is why they can produce different results. A thorough review is beyond the scope of the present manuscript, but a reference is given to Butler et al. (2018), where such a review is presented.

**line 115: There is also a nice table in recently published work by Mertens et al. (2020) (https://doi.org/10.5194/acp-20-7843-2020) discussing in detail the characteristics of these two methods, which might be helpful here.**

Table 1 from Mertens et al. (2020) does make a nice contribution to the discussion of the complementary nature of tagging and perturbation approaches. We have added a reference.

**line 115: Perhaps you want to adapt the naming consistently throughout the manuscript? contribution/share for tagging approaches changes/impact for perturbation? E.g. line 56 Hoor et al used perturbation approaches. The wording "contribution" in this respect might be misguiding.**

265 We agree that the use of consistent language is important. As well as the discussion of Hoor et al. (2009) in the introduction, we have also modified the discussion of Jonson et al. (2020) in Section 3.2.2 to avoid using the term "contribution" to describe results obtained with perturbation approaches.

**line 126: This statement might be misunderstood. The Butler et al tagging scheme is the only one, which separately analyses attribution of tropospheric ozone to its NOx and reactive carbon precursors, whereas**
270 **the Grewe et al scheme is the only, which analysis attribution of tropospheric ozone to both together, NOx and reactive carbon precursors, taking competing effects into account.**

This comment is related to an earlier comment by the same referee regarding the nature of ozone source attribution. In addition to our earlier response, we have added another sentence to the manuscript here as well to re-emphasise the point that there are different approaches to ozone source attribution. We also note that the original manuscript already contains a reference to
275 Grewe et al. (2017) as an example of ozone source attribution with tagging.

Since the referee has opened the discussion on different tagging approaches, we take the opportunity here to also comment on the recent work of Bates and Jacob (2020). Given the diversity of tagging approaches, for ozone we believe that an inter-comparison of these different approaches could be informative. We have added some text mentioning this to the paragraph in question, as well as to the end of Section 3.1.2, where we had already called for the widespread implementation of tagging
280 techniques into models for use in model inter-comparisons.

**line 159: Is there any reason why the spin-up differs?**

Yes, the reason for the longer spinup in the reactive carbon run is because the tag representing ozone produced from methane takes longer to spin up than any of the ozone tags in the $NO_x$ run, consistent with the longer lifetime of methane. The other referee also asked for more detail about the spinup, and we have expanded the discussion here accordingly.

285 **line 233: also Dahlmann et al. 2011.**

We have added the reference.

**line 401: Please elaborate a little bit more on this comparison in terms of quantitative values. I guess we should expect a difference in the strength of the contribution vs perturbation? If so, can that be explained by the difference in the method?**

290 We do not believe that it is currently possible to perform an apples-to-apples comparison of contributions to surface ozone derived from perturbation studies as reported in the literature, with the contributions calculated here through tagging. We agree with previous work from the referee that these approaches can often yield different results (Grewe et al., 2010; Mertens et al., 2018). Given the other differences between our work and the previous literature on perturbation studies (different models, different emissions, different years, etc...) we tend to believe that any quantitative comparison could be misleading, so we
295 prefer to stick with our cautious, qualitative comparisons.

The authors believe that the best way forward here is to carefully design future studies so that the source attribution methods themselves can be compared, rather than any other confounding factors which may be present. The work by Mertens et al. (2018) is a good example of a study which combines both perturbation and tagging in a consistent framework. We also believe that our own methane perturbation work in the present manuscript is also useful in this respect.

300 **line 550: RF and human health effects are not calculated. Please re-phrase that this is a potential important impact based on literature and not your findings**

We have modified the text to indicate that this is prior knowledge.

**Figure 4: Please adapt the text in the figure to explicitly state that surface ozone is presented. Caption: Please include some more details, e.g. "Source-receptor relationships between annual averaged surface ozone vol-**
305 **ume mixing ratio and NOx and VOC emission type and region.", in order to clarify that with region the NOx emission and not the ozone production is meant.**

We agree that the key word "surface" was missing from our caption, so we have included this. We find the referee's suggested rewording of the caption somewhat awkward, so we have attempted to rewrite the caption along the lines suggested by referee.

**References**

[revised manuscript text omitted]

360    number D11 305, https://doi.org/10.1029/2005JD006605, 2006.